

**Effect of straw retention and mineral fertilization on P**
**speciation and P-transformation microorganisms in water**
**extractable colloids of a Vertisol**
Shanshan Bai [a,b], Yifei Ge[a], Dongtan Yao[a], Yifan Wang[a], Jinfang Tan[a,b], Shuai Zhang[c], Yutao Peng[a],
Xiaoqian Jiang [a,b*]
*[a] School of Agriculture and Biotechnology, Sun Yat-sen University, Guangzhou, Guangdong 510275, PR China*
*[b] Modern Agricultural Innovation Center, Henan Institute of Sun Yat-sen University, Zhumadian, Henan 463000, PR*
*China*
*[c] Beijing Key Laboratory of Farmland Soil Pollution Prevention-control and Remediation, College of Resources and*
*Environmental Sciences, China Agricultural University, No. 2 Yuanmingyuan Xilu, Haidian, Beijing 100193, PR*
*China*
* Corresponding author: jiangxq7@mail.sysu.edu.cn (X. Jiang).

## Abstract

Water extractable colloids (WECs) serve as crucial micro particulate components in soils, playing a vital
role in the cycling and potential bioavailability of soil phosphorus (P). Yet, the underlying information
regarding soil P species and P-transformation microorganisms at the microparticle scale under long-term
straw retention and mineral fertilization is barely known. Here, a fixed field experiment (~13 years) in a



Vertisol was performed to explore the impacts of straw retention and mineral fertilization on inorganic P,
organic P and P-transformation microorganisms in bulk soils and WECs by sequential extraction
procedure, P K-edge X-ray absorptions near-edge structure (XANES), $^{31}$P nuclear magnetic resonance
(NMR), and metagenomics analysis. In bulk soil, mineral fertilization led to increases in the levels of
total P, available P, acid phosphatase (ACP), high-activity inorganic P fractions (Ca$_2$-P, Ca$_8$-P, Al-P, and
Fe-P) and organic P (orthophosphate monoesters and orthophosphate diesters), but significantly
decreased the abundances of P cycling genes including P mineralization, P-starvation response regulation,
P-uptake and transport by decreasing soil pH and increasing P in bulk soil. Straw retention had no
significant effects on P species and P-transformation microorganisms in bulk soils but brought increases
for organic carbon, total P, available P concentrations in WECs. Furthermore, straw retention caused
greater change in P cycling genes between WECs and bulk soils compared with the effect of mineral
fertilization. The abundances of *phoD* gene and *phoD*-harbouring *Proteobacteria* in WECs increased
significantly under straw retention, suggesting that the P mineralizing capacity increased. Thus, straw
retention could potentially accelerate the turnover, mobility and availability of P by increasing the
nutrient contents and P mineralizing capacity in microscopic colloidal scale.
**Keywords**: water extractable colloids, inorganic P, organic P, P-cycling genes, straw retention, mineral
fertilization



## 1. Introduction

Phosphorus (P) has a vital function in the productivity of agroecological system (Jiang et al., 2015).

Vertisol (Staff, 2010), also known as a Shajiang black soil in Chinese Soil Taxonomy, covers

approximately $4 \times 10^6$ hectares in the Huang-Huai-Hai Plain of China (Guo et al., 2022). The

characteristics of the Vertisol contain abundant calcium, scant organic matter, and poor fertility (Chen et

al., 2020). The strong P fixation capacity by abundant calcium and poor supply capacity of P restrict

agricultural production severely (Ma et al., 2019). Straw retention and mineral fertilization are commonly

employed to enhance soil nutrient contents in this area (Zhao et al., 2018). Under mineral fertilization

and straw retention, $Ca_2$-P, Fe-P and Al-P contents increased, but $Ca_{10}$-P concentration reduced, thereby

promoting the transformation of P fractions (Xu et al., 2022). Cao et al. (2022) suggested that the

combination of straw retention and mineral fertilization significantly increased both inorganic and

organic P species concentrations. Crop straw, which is rich in organic matter and contains a certain

amount of nitrogen (N), P, and other nutrients, has demonstrated potential effects on the cycling and

processing of P (Damon et al., 2014).

The assessment of potential bioavailability and mobility of soil P heavily relies on the speciation and

distribution of P in soil aggregates (Ranatunga et al., 2013). Agricultural management practices like the

application of fertilizer and straw could modify the microhabitat's physicochemical environment through

their influence on soil aggregation (Ju et al., 2023). Maize straw promoted the accumulation and



55 stabilization of inorganic and organic P in soil aggregates, particularly in the 250–2000 μm fraction.

56 Additionally, it decreased the relative contribution rates of the <53 μm fraction to inorganic and organic

57 P fractions compared with mineral fertilizer (Cao et al., 2021). Generally, soil aggregate fractionation

58 contains the particle size of > 0.25 mm, 0.053-0.25 mm, and <0.053 mm, and the distribution and

59 dynamics of P in these aggregates have been widely researched (Cheng et al., 2019; Deng et al., 2021).

60 However, there are few studies on the forms and distribution of P in soil water-extractable colloids

61 (WECs; <2 μm in size), which significantly contribute to P cycling due to the large binding ability, high

62 mobility and bioavailability of P (Fresne et al., 2022; Jiang et al., 2023). WECs, readily extracted upon

63 water contact, are regarded as indexes of mobile soil colloids (Missong et al., 2018) and main factors

64 that impact the mobility and availability of soil P (Zhang et al., 2021). Colloidal P could contribute to

65 plant-available P as reported by Montavo et al. (2015). Additionally, the microaggregates (including

66 colloidal size fractions) provided a favorable habitat for microorganisms and the biochemical processes

67 functioning at the microparticle scale would be also important for soil P cycling and availability (Totsche

68 et al., 2018). However, the information related to how straw retention and mineral fertilization

69 managements affect soil P dynamics at scales of WECs remains scarce.

70 Microorganisms are instrumental in facilitating the transformation of soil P species, P cycling and P

71 availability regulation (Bergkemper et al., 2016). The processes of microbial P transformation primarily

72 consists of: (1) inorganic P solubilization (e.g., *gcd*); (2) organic P mineralization (e.g., *phoD, phoA, phy*);



(3) P starvation response regulation (e.g., *phoR*, *phoB*); and (4) P uptake and transport system (e.g., *pst*)
(Richardson and Simpson, 2011). Fertilization could further change the abundance and taxonomic
assignments of P cycling gene clusters (Dai et al., 2020; Zhang et al., 2023). For example, continuous N
fertilization over an extended period may lead to a decline in soil pH, inhibition of microbial growth,
alterations in the composition of the microbial community, and ultimately the reduction in the capacity
for P solubilization (Rousk et al., 2010). Additionally, genes expression related to organic P
mineralization, P-starvation regulation, P-uptake and transport are primarily affected by the
environmental P supply (Hsieh and Wanner, 2010). Several researches have shown that the adequate P
supply inhibited the genes expression associated with P-starvation response (e.g. *phoR*), as well as genes
encoding alkaline phosphatase (e.g. *phoD*) and phytase (e.g. *phy*) (Yao et al., 2018; Xie et al., 2020).
Straw retention could bring the increase in soil organic C, potentially enhancing the diversity and richness
of *phoD*-harboring microbes and the *phoD* abundance (Cao et al., 2022). Moreover, alterations in the P
transformation genes are driven by the structural effects of soil aggregates in addition to P availability
(Neal et al., 2017). However, little is known about the richness and distribution of genes related to P
transformation in WECs fraction with the treatments of straw retention and mineral fertilization, which
will offer a new perspective on P cycling and availability from a microbial perspective.
The long-term field experiments (~13 years) under straw retention and mineral fertilization were
conducted. This study aims to: (1) investigate the responses of P speciation, P-cycling-related genes and



taxonomic assignments in bulk soils and WECs under straw retention and fertilization management
strategies; (2) explore the relationship between P species, P-transformation genes and soil properties.
Finally, these results could elucidate the underlying mechanisms of soil P cycling and availability under
mineral fertilization and straw retention from the microparticle and microbial perspective, providing an
important insight into regulating P cycling in agriculture soils.

## 2. Materials and methods


### 2.1 Experimental design


In 2008, a field trial was conducted in Mengcheng County (33°9′N, 116°32′E), Anhui Province, China,
to investigate the rotation of winter wheat and summer maize. The soil is classified as a Vertisol (Staff,
2010), which is derived from fluvio-lacustrine sediments. The region experiences an average annual
temperature and precipitation of 14.8℃ and 732.6 mm respectively.
Six treatments with three replicates (each plot area was 43.2 $m^2$) were carried out: (1) the control
treatment, without straw retention and mineral fertilizer (W0M0F0), (2) single application of mineral
fertilizer (W0M0F1), (3) maize straw retention combined with mineral fertilization (W0M1F1), (4)
wheat straw retention combined with mineral fertilization (W1M0F1), (5) both wheat and maize straw
retention without fertilization (W1M1F0), and (6) a combination of both wheat and maize straw retention
with mineral fertilization (W1M1F1). In the W0M1F1 treatment, maize straw was chopped into
fragments approximately 10 cm in length and uniformly distributed in each plot after harvest, while



wheat straw was removed. In the W1M0F1 treatment, wheat straw was similarly returned to plots and
maize straw was removed. For W1M1F0 and W1M1F1 treatments, maize and wheat straw are both
returned to plots when they are harvested. The amounts of residue incorporation for wheat and maize
were 7500 and 12000 kg/ha respectively. For the fertilization treatments (i.e., W0M0F1, W0M1F1,
W1M0F1, W1M1F1), 240.0 kg/ha N (55% as basal fertilizer and 45% as topdressing during the reviving-
jointing period), 90.0 kg/ha P, and 90.0 kg/ha K (100% as basal fertilizer) were applied in each growing
season of winter wheat. The 300.0 kg/ha N (50% as basal fertilizer and 50% as topdressing at the flare
opening period), 90.0 kg/ha P and 90.0 kg/ha K (100% as basal fertilizer) were applied in each growing
season of summer maize. The fertilizers comprised of compound and urea fertilizer (N-$P_2O_5$-$K_2O$: 15-
15-15). The contents of P in maize straw and wheat straw was about 1.5 and 0.8 g/kg respectively (Chai
et al., 2021). In addition, weeds, disease, and pest control for both wheat and maize were consistent.
**2.2 Soil sampling and water extractable colloids (WECs)**
The soil samples with six treatments were conducted after wheat harvest in June 2021. Five soil cores
(0–20 cm) were gathered from each replicate plot using the quincunx sampling method, and then blended
evenly to create a composite sample. The divisions of three subsamples were made for each sample. The
first subsample was preserved at 4 ℃ to examine P (MBP) and microbial biomass C (MBC), along with
the acid and alkaline phosphatase activities (ACP and ALP). Another sample was at stored −80 °C for
metagenomics analysis. For other soil chemical properties test, the last sample was subjected to air-



drying, grinding, and subsequently sieving through a 2 mm mesh. In this study, the soil fraction consisting
of particles smaller than 2 mm was designated as bulk soil.
To further investigate the impact the sole straw retention and sole mineral fertilization on P cycling in
soil colloids, the particle-size fractionation method following Stokes' Law (Sequaris and Lewandowski,
2003) was utilized to obtain WECs for the W0M0F0, W0M0F1 and W1M1F0 treatments in this study.
About 113-116 g of moist soil samples (equivalent to 100 g of dry soil) was blended with 200 mL
ultrapure water, and then shaken at a speed of 150 rpm for a duration of 6 h. Afterward, we added an
extra 600 mL of ultrapure water and blended thoroughly. The particles >20 μm were allowed to settle for
a period of 6 min. The 2-20 μm was then obtained by eliminating the supernatant following an addition
sedimentation of 12 h. The final supernatant containing colloidal particle fraction (<2 μm) was obtained
and defined as WECs. The proportions of particles with >20 μm, 2-20 μm and <2 μm to bulk soil were
shown in Fig. S1.
**2.3 Soil chemical properties**
A pH meter was utilized to measure soil pH. An elementary analyzer was utilized for soil organic carbon
(SOC), and total nitrogen (TN). After microwave digestion, total P concentrations (TP) were gained by
inductively coupled plasma optical emission spectroscopy (ICP-OES). Available P (AP, Olsen-P)
concentration was quantified by Olsen and Sommers (1982).
The chloroform fumigation method outlined by Vance et al. (1987) and Brookes et al. (1982) was utilized



to quantify the soil MBC and MBP. The extracted C with 0.5 M $K_2SO_4$ in non-fumigated and fumigated
samples was determined with the Multi N/C 2100S TOC-TN analyzer. The dissolved organic carbon
(DOC) was quantified as the extracted organic C by $K_2SO_4$ extract from the non-fumigated samples (Wu
et al., 2019). MBC was quantified by measuring the variation in extractable C content between the non-
fumigated and fumigated soil samples, using the universal conversion factor of 0.45. MBP was calculated
as the variation in extractable P with 0.5 M $NaHCO_3$ between the non-fumigated and fumigated soil
samples, with a conversion factor of 0.40. The measurement of ACP and ALP followed the procedures
outlined by Tabatabai and Bremner (1969).
**2.4 Phosphorus sequential extraction procedure and P K-edge XANES spectroscopy**
The modified sequential extraction procedure, as described by Jiang and Gu (1989) and Audette et al.
(2016), was utilized to extract various P fractions including $Ca_2$-P, $Ca_8$-P, Al-P, Fe-P, occluded-P (O-P)
and $Ca_{10}$-P in bulk soils. Then the method outlined by Murphy and Riley (1962) was utilized to ascertain
the concentration of each P fraction.
P K-edge X-ray absorptions near-edge structure (XANES) spectra were utilized to clarify the P bonding
fractions in WECs, and acquired at Beamline 4B7A of the Beijing Synchrotron Radiation Facility,
Beijing, China. Dibasic calcium phosphate dihydrate (DCP, $CaHPO_4 \cdot 2H_2O$), hydroxyapatite (HAP,
$Ca_5(PO_4)_3OH$), aluminum phosphate (Al-P, $AlPO_4$), iron phosphate dihydrate (Fe-P, $FePO_4 \cdot 2H_2O$) and
inositol hexakisphosphate (IHP) were chosen as references. For XANES data collection, P references



and soil samples were thinly spread on the carbon tape with a P-free, double-sided in PFY mode with a
SiLi detector. Multiple spectra were obtained with three duplicates for each sample and then averaged.
The spectra were studied using Athena (0.9.26) with the energy calibration at 2149 eV (E0), aligning
with the peak position of $AlPO_4$, as described by Beauchemin et al. (2003). Then, we performed the
Linear combination fitting (LCF) within the energy range spanning from −10 eV to 30 eV relative to E0,
and the goodness of fit was determined based on the chi-squared and R values. The most likely P species
considered was considered based on these results. The P K-edge XANES spectra of P reference
compounds were as shown in Fig. S2.
**2.5 Solution $^{31}$P NMR spectroscopy**
Solution $^{31}$P-NMR spectroscopy were performed to clarify P species (Turner, 2008). The 1 g bulk soil
and WECs sample was mixed with 10 mL of 0.25 M NaOH and 0.05 M $Na_2EDTA$ and shaken for 4 h to
extract P (Cade-Menun and Liu, 2014; Jiang et al., 2017). The procedure was outlined in our prior study
(Bai et al., 2023). The $^{31}$P-NMR spectra were acquired using a Bruker 500-MHz spectrometer with 4.32
s relaxation delay, 0.68 s acquisition time, 5000 scans, and 90° pulse width(Cade-Menun et al., 2010).
Compound identification relied on their chemical shifts following the calibration of the orthophosphate
peak to 6.0 ppm (Table S1). To validate peak identification, samples were spiked with *myo*-inositol
hexakisphosphate, α- and β- glycerophosphate, as well as adenosine monophosphate (Fig. S3). Instead
of being classified as monoesters, the α- and β-glycerophosphate as well as mononucleotides (Glyc+nucl)



were categorized as orthophosphate diesters (Doolette et al., 2009). Integration was conducted on spectra
with broadening at 7 and 2 Hz to calculate the area under each peak. To quantify the concentrations of P
species, the peak areas were multiplied by the concentration of NaOH-Na$_2$EDTA extractable P. The
spectra of bulk soil and WECs were processed using MestReNova 10.0.2 software, as shown in Fig. S4.
**2.6 DNA extraction and metagenomics analysis**
The process of soil DNA extraction was carried out with a FastDNA Spin kit (MP Biomedicals, USA).
The Agilent 5400 was utilized to determine the purity, integrity and concentration of the extracted DNA.
The generation of sequencing libraries was carried out using the NEBNext® Ultra™ DNA Library Prep
Kit (PerkinElmer, USA). For each sample, barcodes were incorporated to enable sequence attribution.
After end-polished, A- tailing, and adapter ligation, the DNA fragments were subsequently subjected to
PCR amplification. Finally, a NovaSeq 6000 instrument was utilized for sequencing, generating paired-
end reads. Reads containing low-quality bases and N base were removed (Hua et al., 2015).
MEGAHIT was used to assemble the filtered reads (fastq formats) by *de Bruijn* graph with the minimum
k-mer size of 21 (Li et al., 2015). The default settings of Prodigal were used to identify the protein-
coding genes, as described by Hyatt et al. (2010). For functional annotation, we employed the Diamond
software to align the identified genes against the nonredundant protein sequences database of NCBI and
Kyoto Encyclopedia of Genes and Genomes (KEGG) databases following the methodologies as outlined
by Kanehisa and Goto (2000), Buchfink et al. (2015) and Huson et al. (2016).





According to the prior studies of Bergkemper et al. (2016), a cumulative of 29 genes associated with P-
transformation were identified, along with their corresponding KO numbers. These genes were
categorized into four distinct groups: (1) genes associated with inorganic P-solubilization; (2) genes
associated with organic P-mineralization; (3) genes associated with P-starvation regulation, and (4) genes
associated with microbial P-uptake and transport. Table S2 provides a comprehensive list of the
categorized genes along with their names, function descriptions, and KEGG Orthology (KO) numbers.
The sequence data have been submitted in the NCBI Sequence Read Archive (PRJNA909638).
**2.7 Statistical analysis**
The IBM SPSS and R software were utilized for statistical analyses and data visualization. The normality
distribution (Shapiro–Wilks tests) were performed before ANOVA. To identify significant differences
among mean values at a significance level of 0.05, the Tukey's honestly significant differences (HSD)
test was employed. The differences of soil properties, total P, inorganic P, organic P, ACP, and ALP
between bulk soils and WECs were tested by independent-samples T test. The differences of P cycling
genes composition in bulk soils and WECs were displayed by principal component analysis (PCA).
Principal coordinate analysis (PCoA) was utilized to present the microbial bacterial β-diversity for
typical P-solubilization (*gcd*) and mineralization (*phoD*) genes. The associations between the abundances
of P-transformation genes and soil characteristics were assessed using Spearman's correlations with the
correlation coefficients ($R$) > 0.6 and P-value <0.05. Structural equation modeling (SEM) was used to



explore the relationships among agricultural managements, soil properties, and P-cycling related genes
by Amos (24.0). The model fit was assessed with goodness of fit (GFI) and root square mean error of
approximation (RMSEA).

## 3. Results

### 3.1 Soil properties in bulk soils and WECs

Straw retention incorporated with mineral fertilization (i.e., W0M1F1, W1M0F1, W1M1F1) decreased
soil pH by 1.76-1.89 units and alkaline phosphatase activity (ALP) by 160.25-183.37 μg/(g·h)
significantly, but increased significantly organic C by 2.66-4.73 g/kg, total N by 0.36-0.60 g/kg, total P
by 0.17-0.19 g/kg, available P by 28.11-31.97 mg/kg, and acid phosphatase activity (ACP) by 174.12-
449.25 μg/(g·h), respectively compared with the control treatment (i.e., W0M0F0) (Table 1). The
variations primarily resulted from the utilization of mineral fertilizers, as there were no noteworthy
distinctions observed in these parameters between straw retention combined with mineral fertilization
treatments and sole mineral fertilizer (i.e., W0M0F1). The application of straw retention (i.e., W1M1F0)
had little effect on these soil properties except for slight increases in soil MBC and MBP contents
compared with the control treatment. The outcomes suggested mineral fertilization showed more
prominent impact on soil characteristics compared to that of straw retention. Mineral fertilization indeed
enhanced soil nutrient contents, but caused soil acidification. The soil acidification was not effectively
alleviated under straw returning combined with mineral fertilization.



The WECs accounted for 9.73-11.05% of bulk soils, and the proportions of WECs were not affected by
mineral fertilization and straw retention (Fig. S1). The significantly higher concentrations of SOC, TN,
TP and available P were monitored in WECs than those in bulk soils for all the tested samples including
the control treatment (i.e., W0M0F0), sole mineral fertilization (i.e., W0M0F1) and sole straw retention
(i.e., W1M1F0) (Fig. 1 A-D). The influence of either mineral fertilization or straw retention on
physicochemical properties of WECs was more obvious than that on bulk soils. For example, organic C
and total N contents in WECs experienced a substantial rise following the implementation of straw
retention compared with the control treatment from Fig. 1 A and B.
**3.2 P bonding fractions in bulk soils and WECs**
The concentrations of total inorganic P and $Ca_2$-P, $Ca_8$-P, Al-P, and Fe-P under straw retention
incorporated with mineral fertilization increased remarkably by 128.93-146.99 mg/kg, 15.41-17.30
mg/kg, 3.19-4.38 mg/kg, 59.74-68.97 mg/kg, and 44.08-54.46 mg/kg, respectively compared with the
control as shown in Table 2. Accordingly, the marked increases in the proportion of $Ca_2$-P, $Ca_8$-P, Al-P,
and Fe-P were observed, while the proportion of $Ca_{10}$-P decreased remarkably (Fig. S4). These
differences were mainly caused by mineral fertilization. There was also no significant difference between
straw retention incorporated with mineral fertilization and sole mineral fertilization. The straw retention
had little impact on the concentrations of each inorganic P fraction compared with the control.
According to the XANES analysis of WECs, there were notable increases in the proportions of Al-P and



Fe-P, but remarkable decreases in the proportions of DCP and IHP was observed after mineral
fertilization compared with the control treatment (Table 3 and Fig. S5). However, the straw retention
brought slight increases in the proportions of Fe-P and IHP.
**3.3 Solution $^{31}$P NMR analysis of bulk soils and WECs**
The concentrations and proportion of orthophosphate in bulk soils increased by 146.4-182.6 mg/kg and
18.6-21.3% significantly under straw retention incorporated with mineral fertilization compared with the
control and sole straw retention treatments (Table 4 and Fig. S6A). Organic P concentrations also
increased under mineral fertilization, among which orthophosphate monoesters and orthophosphate
diesters increased by 12.78-27.00 mg/kg and 7.55-10.05 mg/kg, respectively. Furthermore, the
concentration of each P specie in bulk soil showed no notable difference between straw retention
incorporated with mineral fertilization treatments and sole mineral fertilization treatment (Table 4). In
comparison with the control, the concentration of orthophosphate monoesters and orthophosphate
diesters in bulk soil increased slightly under sole straw retention, but this difference was not statistically
significant. These results manifested that the effect of mineral fertilization on P species concentration
was more apparent than that of straw retention.
Notably, the concentrations of orthophosphate, orthophosphate monoesters, orthophosphate diesters, and
Glyc+nucl (i.e., α/β-glycerophosphate and mononucleotides) in WECs were significantly greater (~2.5
times) than those in bulk soil for all the tested samples (Table 4 and 5). Mineral fertilization had more



significant effects on the concentrations of P species in WECs compared with those in bulk soils. Relative
to the control, the concentrations of orthophosphate, orthophosphate monoesters and orthophosphate
diesters rise sharply after mineral fertilization for WECs, while the significant increase of only
orthophosphate concentrations was detected for bulk soils. Furthermore, the concentration of these P
species in WECs under sole straw retention increased slightly in comparison with the control (Table 5).
**3.4 Genes associated with P transformation in bulk soils and WECs**
In bulk soils, there were remarkable decreases in total relative abundances of genes associated with P-
transformation under the combined application of straw retention and mineral fertilization compared with
the control. These genes included those related to organic P-mineralization (e.g., *phoA*, *phoD*, *phy*, *ugpQ*),
P-starvation regulation (e.g., *phoR*), P-uptake and transport (e.g., *phnCDE*) as described in Figs. 2A and
B. No notable difference was observed in the abundances of these P transformation genes in bulk soils
between straw retention combined with mineral fertilization and sole mineral fertilization, but they were
significantly different from those for sole straw retention. This indicated that the decrease in abundances
of P transformation genes was mainly caused by mineral fertilization but not by straw retention.
Correspondingly, the PCA results also revealed clear separations for the genes related to P-cycling
between with (i.e., W0M0F1, W1M0F1, W0M1F1, and W1M1F1) and without (i.e., W0M0F0 and
WM1F0) mineral fertilization treatments (Fig. 3 A).
The PCA analysis (Fig. 3 B) exhibited a clear segregation between the P-cycling genes in WECs and



those in bulk soils for all the tested samples, including sole mineral fertilization, sole straw retention and
the control treatments. Straw retention caused significant differences of relative abundance for many
gene species including *ppa*, *ppk*, *phoD*, *phoN*, *phy*, *phoR*, *phnCDE* and *ugpBAEC* between WECs and
bulk soils. In contrast, sole mineral fertilization caused significant differences of less gene species
including *gcd*, *ppx*, *glpABCK* and *phoR*, and the control treatment caused significant differences of
*glpABCK* and *phoR* genes (Fig. 4 B). These results suggested that straw retention caused greater change
of P cycling gene between WECs and bulk soils compared with mineral fertilization.
**3.5 Taxonomic assignments of *phoD* and *gcd* genes**
The *phoD* gene (encoding alkaline phosphatases) and *gcd* gene (encoding glucose dehydrogenase for
synthesizing) serve as critical indicators of P mineralization and solubilization, respectively. As shown
in Fig. 4, straw retention caused significant increase of the abundance for *phoD* gene and mineral
fertilization caused significant decrease of the abundance for *gcd* genes in WECs compared with bulk
soils. Thus, we further performed the taxonomic assignments of *phoD* and *gcd* genes.
For bacterial taxa containing the *phoD* gene in WECs (Fig. 5 A), the abundance of *Proteobacteria*
increased significantly under sole straw retention when compared to those in bulk soils. For bacterial
taxa containing the *gcd* gene in WECs (Fig. 5 B), the abundance of *Acidobacteria* decreased significantly
compared with those in bulk soils under mineral fertilization. Additionally, the bacterial β-diversity in
WECs showed a clear divergence from those in bulk soils for all the treatments (Fig. S7).



**3.6 Correlations between P-cycling genes and soil properties, P species in bulk soils and WECs**
According to Spearman's Rank correlations (Fig. S8), more P gene species were correlated with soil
properties and nutrients in bulk soils than WECs (R > 0.6, P <0.05), suggesting that the response of P
cycling genes to soil properties in bulk soil were more sensitive than those in WECs. Specially, a
correlation was detected between the majority of P cycling genes and soil nutrients including C, N, P in
bulk soils. Whereas, there was no consistent trends in WECs.
According to Fig. 6, mineral fertilization influenced the P-cycling genes by decreasing soil pH and
increasing total P in bulk soil. The model fit in bulk soil was : GFI=0.939, RMSEA=0.036. Furthermore,
the decrease in soil pH affected positively the genes involved in organic P mineralization (0.82, P < 0.01)
and the increase in total P had negative effect on the genes involved in P-starvation regulation (-0.77, P
< 0.01). In WECs, agricultural managements affected the P-cycling genes by increasing total P (0.98, P
< 0.01) and organic C (0.92, P < 0.01).The model fit in WECs was : GFI=0.964, RMSEA=0.000.
Moreover, total P had negatively affected the genes related to and organic P mineralization (-0.67, P <
0.01) and inorganic P solubilization (-0.69, P < 0.05).

## 4. Discussions


**4.1 Response of soil properties, P species and transformation genes in bulk soils**
In bulk soil, mineral fertilization decreased soil pH, increased soil TP, thus decreasing the abundances of
P transformation genes. Soil acidification might be due to the increased protons release from nitrification



processes occurring under mineral N fertilization (Guo et al., 2010). The significant increases in TP
concentrations under mineral fertilization might be closely associated to the enhanced organic matter
from crop residues and the input of P fertilizers (Zhang et al., 2018). Moreover, Tong et al. (2019)
reported that mineral fertilization also increased root exudates, which brought the increases in soil
organic matter and nutrients.
Generally, the P mineralization, P-starvation regulation, P-uptake and transport genes were primarily
influenced by the environmental availability of P (Hsieh and Wanner, 2010; Richardson and Simpson,
2011). Under conditions of low soil P, microorganisms exhibited an upregulation of genes within the *Pho*
regulon, specifically those encoding phosphatases and phosphate transporters (Vershinina and
Znamenskaya, 2002). The expression of *phoR* and *phoD* was governed by the presence of P starvation
conditions (Xie et al., 2020). The phytase was inhibited by high level of phosphate (Yao et al., 2018) and
higher abundance of *phy (3-phytase)* was observed in P-deficient soils compared to P-rich soils (Siles et
al., 2022). The *ugpQ* gene also usually accumulated in P starvation conditions as the operon of
glycerophosphodiester-utilizing system (Luo et al., 2009). Therefore, in the control and straw retention
treatments with lower P concentrations, higher abundances of *phoD*, *phy*, *phoR*, and *ugpQ* genes were
observed in comparison with the mineral fertilization treatments (Fig. 2). Mineral fertilization reduced
the abundance of genes about P mineralization (e.g., *phoA*, *phoD*, *phy*, *ugpQ*), P-starvation regulation
(e.g., *phoR*), P-uptake and transport (e.g., *phnCDE*) significantly (Fig. 2). Consistent with our findings,



prior research has indicated that a notable decline in the *phoD* gene abundance with mineral fertilization
alone or combined with maize straw compared with the control (Ikoyi et al., 2018). Long-term P
application resulted in a reduction in the abundances of *phoR* gene according to Dai et al. (2020).
Additionally, observed changes in soil pH significantly impacted microbial abundances and communities
(Neal et al., 2017; Wan et al., 2021). According to Chen et al. (2017), soil pH was identified as the primary
factor exerting an influence on the microbial community compositions harboring the *phoD* gene, with a
positive correlation observed between the soil pH and the abundance of the *phoD* gene. Studies have
provided evidence that a decrease in soil pH could inhibit bacterial/fungal growth (Li et al., 2020), modify
the microbial community compositions (Rousk et al., 2010), and decrease the relative abundances of
*Actinobacteria* and *Proteobacteria* for *phoD* gene (Luo et al., 2017), which in turn decreases P
mineralization capacity.
According to the Spearman's Rank correlations in this study, the *phoD*, *phoA*, *phy*, *ugpQ*, and *phoR* genes
abundances were correlated negatively with the contents of orthophosphate, orthophosphate monoesters,
orthophosphate diesters, and positively with soil pH ($p<$ 0.05) (Fig. S8 A). Thus, the decline in the
abundance of the P-cycling related genes can be attributed to increasing soil P contents and low soil pH
under mineral fertilization.
In bulk soil, straw retention showed no significant impact on soil properties, P species and transformation
genes. Straw decomposition was affected by the composition of straw (e.g., the C/N, C/P, lignin, cellulose



of straw) and soil characteristics (e.g., soil aeration, pH and nutrient contents). The high C/N, lignin, and
cellulose in wheat and maize straw might slow down straw decomposition (Talbot and Treseder, 2012).
The C/N in wheat and maize straw (52-73:1) were significantly higher than suitable microorganisms C:N
(25-30:1) for straw decomposition (Cai et al., 2018), and microorganisms needed to consume soil original
N when decomposing straw. Therefore, the straw retention without N addition could limit the
decomposition rate of straw. Thus, the straw retention for 13 years did not show any significant impact
on soil C, N, P nutrients. Yet it is noteworthy that although the decomposition rate of straw was slow, it
started to have slight effects on the accumulation of soil microorganisms C and P in bulk soils and was
expected to have a more obvious effect in the longer term. The slow decomposition of straw provided
the nutrients and promoted crop root exudation, consequently fostering the growth of soil microbial and
augmenting soil MBC (Wang et al., 2021). The slight increase in MBC derived the increase of MBP
(Spohn and Kuzyakov, 2013). When N and P fertilizers were added, straw retention incorporated with
mineral fertilization could enhance microbial activity, improve soil microbial C/N and C/P, promote straw
decomposition and increase organic C contents (Li et al., 2018). The input of N and P fertilizers brought
the significant increase in soil N and P contents (Zhang et al., 2018). In this study, straw retention
incorporated with mineral fertilization had remarkable influences on soil characteristics and nutrients,
which was significantly different from sole straw retention. There was no discernible disparity in soil pH
between straw retention incorporated with mineral fertilization and single mineral fertilization, indicating



that straw retention did not alleviate soil acidification caused by mineral fertilization.

**4.2 Response of soil properties, P species and transformation genes in WECs**

The higher concentrations of SOC, TN, TP, AP and P species in WECs compared with bulk soil (Fig. 1)
indicated that nutrients within WECs are enriched, which was because of their high specific surface area
(Jiang et al., 2014). The influences of mineral fertilization and straw retention on soil properties and P
species in WECs were stronger compared with those in bulk soils, suggesting that the physicochemical
properties of soil microparticles were more sensitive than bulk soil in response to soil environmental
disturbance. Soil colloids are the most active constituent, representing the micro particulate phase of soils,
and play a fundamental role in the cycling of P (Fresne et al., 2022). Previous studies demonstrated that
colloids were the important vectors governing P mobility and bioavailability (Rick and Arai, 2011).
According to de Jonge et al. (2004), colloidal P can make a substantial contribution to the transportable
P, amounting to as much as 75% in arable soils. More inorganic and organic P accumulated in the WECs
compared with bulk soils (Tables 4 and 5), which could improve the potential bioavailability and mobility
of P (Krause et al., 2020). Notably, although the practice of straw retention did not result in any significant
changes on nutrient contents in bulk soils, it brought significant increases in TN and SOC contents (Fig.
1 A and B) and slight increases in the concentrations of TP and each P species for WECs. This indicated
that straw retention promoted the accumulation of nutrients on WECs, which exerted a considerable
influence on the supply and cycling of P.



Straw retention caused the greater change of P cycling genes between WECs and bulk soils compared
with mineral fertilization (Fig. 4 B) and led to a significant increase of *phoD* gene in WECs compared
with bulk soils. Research conducted by Fierer et al. (2012) and Ling et al. (2014) suggested that higher
concentrations of total N, P and organic C could favor the growth of microorganisms. For bacterial taxa
containing *phoD* gene, the abundance of *Proteobacteria* (Fig. 5 A) increased significantly in WECs
compared with those in bulk soils under sole straw retention. This indicated that straw retention might
increase the *phoD* gene abundance by influencing *phoD*-harbouring *Proteobacteria*, and then increase P
mineralizing capacity in WECs. Several studies have highlighted that *Proteobacteria* has been
recognized as a crucial group of microorganisms involved in the mineralization of P (Zhang et al., 2023)
and the increase in *phoD*-harbouring *Proteobacteria* could improve potential P mineralization (Xie et al.,
2020). The *Proteobacteria* belongs to copiotrophic microorganisms groups, and accumulates in rich
nutrient soils (Wang et al., 2022). In our research, the notable increases in SOC, TN and each P specie in
WECs were likely to provide favorable conditions of copiotrophic bacteria (e.g., *Proteobacteria*) under
straw retention. Generally, the WECs (clay particles, <2 μm) including natural organic matter (e.g.,
humus) and inorganic colloids (silicate, and Al/Fe oxides) (Zhang et al., 2021) were considered to be the
best natural microorganism adsorbents (Zhao et al., 2014; Madumathi, 2017). Previously conducted
research has indicated that most bacteria (65%) associated with <2 μm soil particulates (Oliver et al.,
2007). The population of the bacteria (*Pseudomonas putida*) attached to the clay particle in Red soil



(*Ultisol*) was significantly higher compared to the populations found on silt and sand particles (Wu et al.,
2012). Furthermore, the increased SOC could improve the surface area and activity of WECs (Zhao et
al., 2014), thus increasing microorganism adhesion (Van Gestel et al., 1996). SOC was a key component
of P binding in colloids (Sun et al., 2023). Thus, we considered that the P cycling microorganisms in soil
colloids might be influenced mainly by the increased nutrients contents.
In this study, although mineral fertilization also caused the enhancements of SOC contents in WECs, it
brought dramatical increase of P contents and decrease of pH by 1.76-1.89 units, which restricted the
abundance of P cycling genes in both WECs and bulk soils as discussed before. Therefore, the difference
of P-cycling genes between WECs and bulk soil under mineral fertilization was less significant than
those under straw retention. Additionally, the consistent change trends of the *gcd* gene and *gcd*-
harbouring *Acidobacteria* indicated that the decrease in *gcd* gene abundance in WECs might be driven
by the *gcd*-harboring *Acidobacteria* under mineral fertilization. (Khan et al., 2007), the *gcd* gene coding
the membrane-bound quinoprotein glucose dehydrogenase (PQQGDH) was involved in the regulation
of the process of making inaccessible mineral P soluble, such as some rock phosphate, hydroxyapatite,
and Ca phosphates. Wu et al. (2021) have shown that the increase in *gcd*-harbouring *Acidobacteria*
improved P solubilization. The *Acidobacteria* was acidophilic and oligotrophic bacteria. Most of their
members lived in low nutrient or high acidity environments. The abundance of *Acidobacteria* was often
negatively correlated with soil nutrient contents and pH (Jones et al., 2009; Rousk et al., 2010). As



mentioned above, soil pH decreased significantly (Table 1) and this might lead to the increase of
*Acidobacteria* in bulk soils after mineral fertilization. The WECs had strong soil buffering capacity by
the exchangeable ion, organic C and clay particles (Curtin and Trolove, 2013; Dvorackova et al., 2022),
and could alleviate the pH change, which did not support the growth of *Acidobacteria*. The pH buffering
capacity and greater nutrient contents in WECs might limit the expression of *Acidobacteria* compared
with bulk soils under mineral fertilization, thus causing the significant decrease in *gcd* gene abundance
in WECs compared with the bulk soil.

## 5. Conclusions

This study provides systematic insights into P speciation and P transformation microorganisms at the soil
microparticle scale (WECs) compared with bulk soil under straw retention and mineral fertilization.
Straw retention caused more obvious impact on the accumulation of organic C and total N of WECs and
the greater change of P cycling genes between WECs and bulk soils even than mineral fertilization. The
significant increase in the abundance of gene encoding for alkaline phosphatase (*phoD*) and *phoD*-
harbouring *Proteobacteria* for WECs compared with bulk soils indicated the improved P mineralization
capacity of WECs under straw retention. This information provided strong evidences that straw retention
could potentially affect the turnover, mobility and availability of P mainly by changing the
physicochemical and biochemical processes involved in the P transformation of soil colloids.

## Acknowledgements



The study was funded by the National Natural Science Foundation of China (No. 42377323) and the
Foundation of Modern Agricultural Innovation Center, Henan Institute of Sun Yat-sen University (No.
N2021-002).
**Declaration of competing interest**
The authors declare no competing interests.
**Supplementary material**
Supplementary material associated with this paper are available on the online version.

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



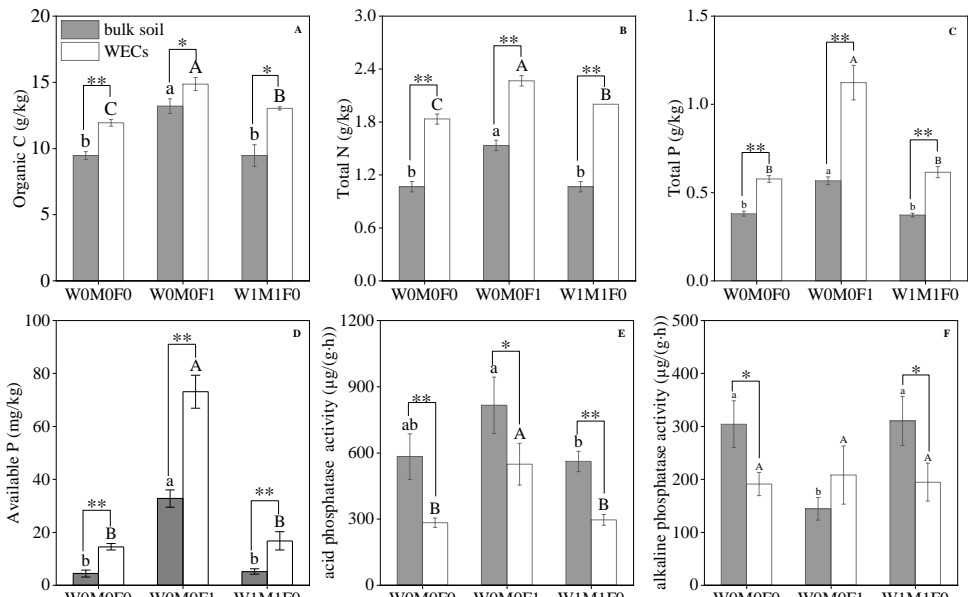

Fig.1 Soil properties in bulk soil and water-extractable colloids (WECs) for the W0M0F0, W0M0F1, W1M1F0

treatments

A: Soil organic carbon (SOC), B: Total nitrogen (N), C: Total phosphorus (P), D: Available phosphorus (P), E: acid phosphatase activity (ACP), F: alkaline phosphatase activity (ALP). Significant differences between treatments in bulk soil are indicated by lowercase letters (p<0.05). Significant differences between treatments in WECs (< 2μm) are indicated by capital letters (p<0.05). Significant differences between bulk soil and WECs are as follows, * p < 0.05 and ** p < 0.01 (Independent-samples T test).



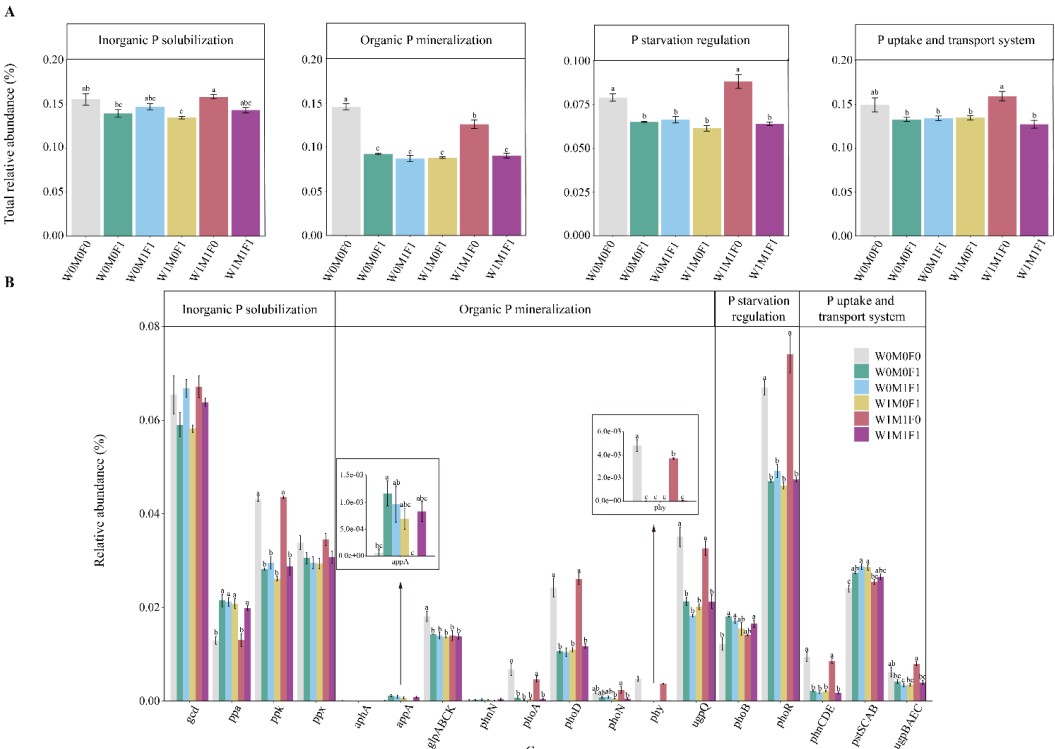

Fig. 2 Relative abundance of representative genes responsible for microbial (1) inorganic P solubilization, (2) organic P-mineralization, (3) P-starvation regulation, and (4) P-uptake and transport in bulk soil

The six treatments were: (1) the control treatment, without straw retention and mineral fertilizer (W0M0F0), (2) single application of mineral fertilizer (W0M0F1), (3) maize straw retention combined with mineral fertilization (W0M1F1), (4) wheat straw retention combined with mineral fertilizer (W1M0F1), (5) both wheat and maize straw retention with no fertilizer (W1M1F0), and (6) both wheat and maize straw retention combined with mineral fertilizer (W1M1F1) respectively. The relative abundances of genes were calculated related to the annotated reads. Significant differences between treatments in bulk soil are indicated by lowercase letters (p<0.05). The relative abundance of glp transporter systems was calculated as the average abundances of gene glpA, glpB, glpC, and glpK; the phn transporter systems was calculated as the average abundances of gene phnC, phnD, and phnE; the pst transporter systems was calculated as the average abundances of gene pstS, pstC, pstA, and pstB; The ugp transporter systems was calculated as the average abundances of gene ugpB, ugpA, ugpE, and ugpC.



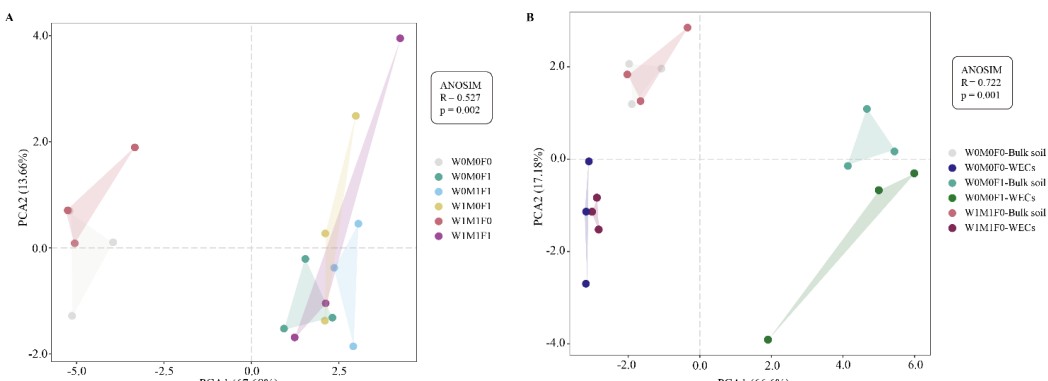

Fig. 3 Principal component analysis (PCA) of P-transformation gene composition in bulk soil (A) and water-extractable

colloids (WECs, B)





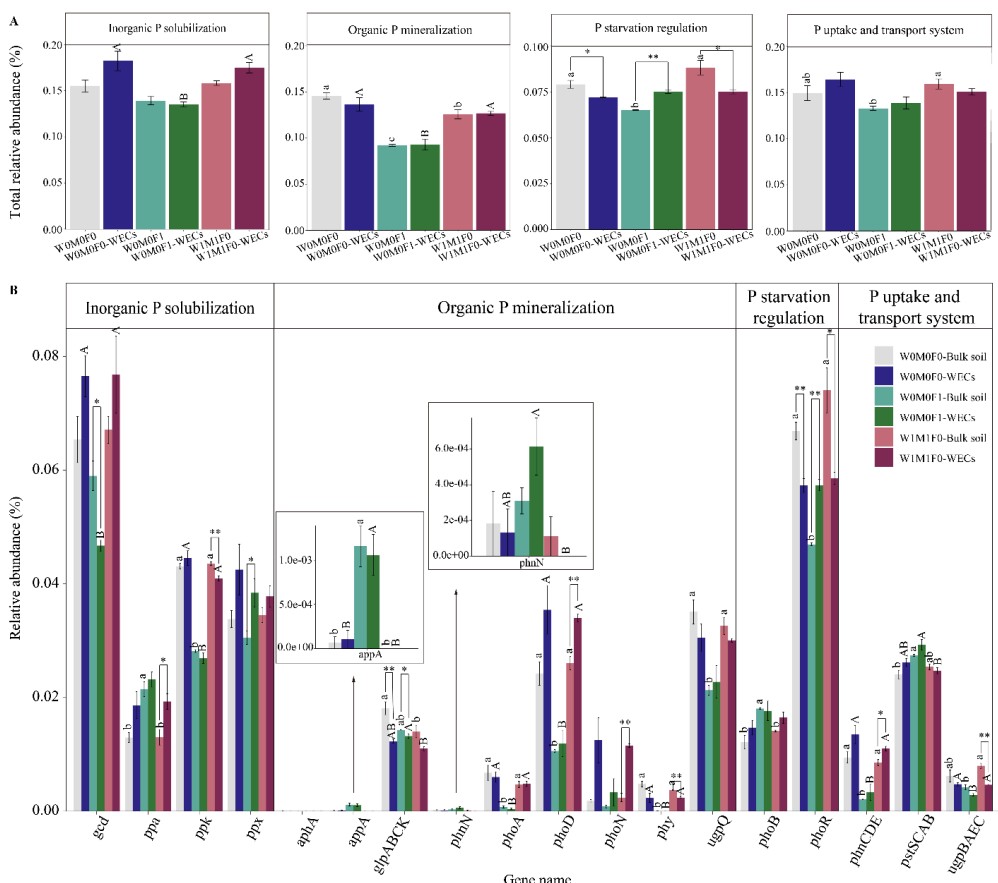

Fig. 4 Relative abundance of representative genes responsible for microbial (1) inorganic P solubilization, (2) organic P-mineralization, (3) P-starvation regulation, and (4) P-uptake and transport in bulk soils and water-extractable colloids (WECs) among the W0M0F0, W0M0F1, and W1M1F0 treatments

The relative abundances of genes were calculated related to the annotated reads. Significant differences between treatments in bulk soil are indicated by lowercase letters ($p < 0.05$). Significant differences between treatments in WECs (< 2μm) are indicated by capital letters ($p < 0.05$). Significant differences between bulk soil and WECs are as follows, * $p < 0.05$ and ** $p < 0.01$ (Independent-samples T test). The relative abundance of glp transporter systems was calculated as the average abundances of gene glpA, glpB, glpC, and glpK; the phn transporter systems was calculated as the average abundances of gene phnC, phnD, and phnE; the pst transporter systems was calculated as the average abundances of gene pstS, pstC, pstA, and pstB; The ugp transporter systems was calculated as the average abundances of gene ugpB, ugpA, ugpE, and ugpC.



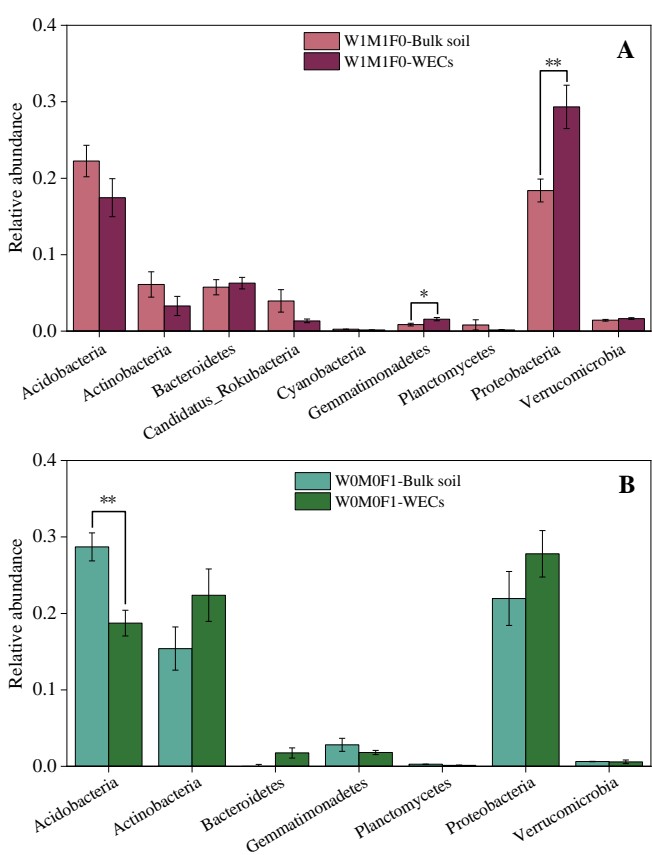

Fig. 5 Taxonomic assignments of phoD gene for the W1M1F0 treatment (A) and gcd gene for the W0M0F1 treatment (B)

at the phylum level in bulk soil and water-extractable colloids (WECs)





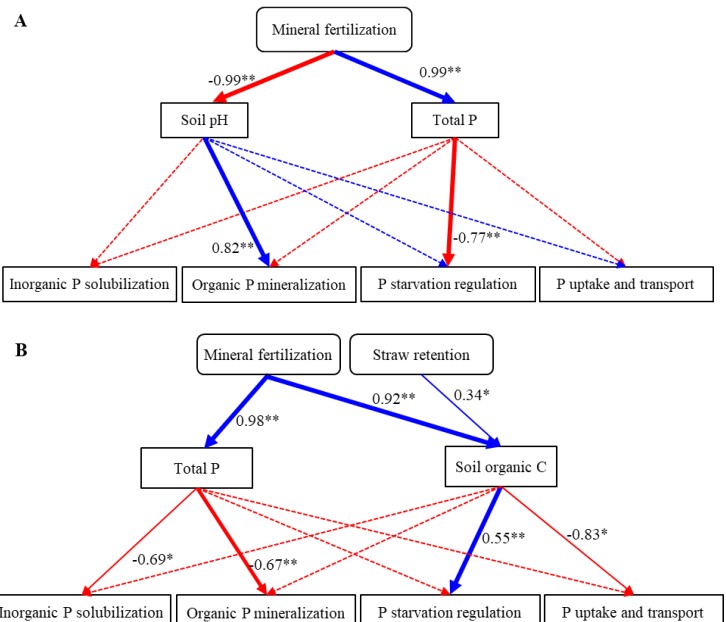

Fig. 6. Structural equation model (SEM) showing the relationship among mineral fertilization and straw retention, soil properties, and P cycling-related gene in bulk soil (A) and water-extractable colloids (WECs, B).

The blue and red solid arrows represent the significant positive and negative relationships between different variables. The dashed arrows represent nonsignificant relationships. The numbers near the blue and red arrows are the path coefficients. * , P < 0.05; * *, P < 0.01.



Table 1 Soil properties of bulk soil among six treatments

| Soil properties | W0M0F0 | W0M0F1 | W0M1F1 | W1M0F1 | W1M1F0 | W1M1F1 |
|---|---|---|---|---|---|---|
| pH | 6.90±0.07a | 5.10±0.14b | 5.06±0.09b | 5.14±0.08b | 6.79±0.08a | 5.01±0.31b |
| Gravimetric moisture (%) | 0.14±0.01a | 0.15±0.01a | 0.14±0.01a | 0.15±0.01a | 0.15±0.02a | 0.15±0.01a |
| Soil organic C (g/kg) | 9.47±0.29c | 13.20±0.56ab | 12.13±0.74b | 13.70±0.56ab | 9.47±0.81c | 14.20±0.96a |
| Total N (g/kg) | 1.07±0.06c | 1.53±0.06ab | 1.43±0.06b | 1.67±0.15a | 1.07±0.06c | 1.57±0.06ab |
| Total P (g/kg) | 0.38±0.01b | 0.57±0.02a | 0.56±0.04a | 0.55±0.03a | 0.37±0.01b | 0.56±0.01a |
| Available P (mg/kg) | 4.43±1.34b | 32.77±3.26a | 32.54±3.18a | 36.40±1.35a | 5.18±1.04b | 32.49±4.12a |
| Microbial biomass P (mg/kg) | 6.80±0.44a | nd | nd | nd | 9.01±4.35a | nd |
| Dissolved organic C (mg/kg) | 54.21±2.56b | 133.43±2.80a | 142.03±8.13a | 134.11±3.97a | 57.01±9.61b | 140.01±9.51a |
| Microbial biomass C (mg/kg) | 316.39±59.52a | 357.95±24.32a | 343.28±90.16a | 307.96±27.45a | 336.23±52.37a | 387.89±21.52a |
| Acid phosphatase activity (μg/(g·h)) | 582.80±103.58c | 815.06±128.42abc | 756.92±142.48bc | 1032.05±149.59ab | 506.63±46.11c | 1102.26±133.11a |
| Alkaline phosphatase activity (μg/(g·h)) | 304.01±43.97a | 144.08±21.39b | 120.64±88.90b | 138.34±12.14b | 310.30±46.22a | 143.76±44.88b |

The six treatments were: (1) the control treatment, without straw retention and mineral fertilizer (W0M0F0), (2) single application of mineral fertilizer (W0M0F1), (3) maize straw retention combined with mineral fertilization (W0M1F1), (4) wheat straw retention combined with mineral fertilizer (W1M0F1), (5) both wheat and maize straw retention with no fertilizer (W1M1F0), and (6) both wheat and maize straw retention combined with mineral fertilizer (W1M1F1) respectively. Values are means ± standard error. The "nd" indicates that the microbial biomass P were not detected. Significant differences between treatments are indicated by the different lowercase letters ($p<0.05$).



Table 2 Concentrations (mg/kg) of inorganic P fractions in bulk soil

| Samples | Ca$_2$-P | Ca$_8$-P | Al-P | Fe-P | O-P | Ca$_{10}$-P | Total inorganic P |
|---|---|---|---|---|---|---|---|
| W0M0F0 | 3.39±0.17b | 1.27±0.22b | 25.14±1.29b | 27.46±3.86b | 37.31±3.02c | 119.95±4.70a | 214.53±2.93c |
| W0M0F1 | 20.39±2.83a | 5.58±0.64a | 90.23±8.03a | 71.54±5.20a | 44.91±2.18abc | 119.04±3.11a | 351.69±14.93a |
| W0M1F1 | 18.80±0.45a | 4.46±1.04a | 84.88±13.86a | 72.13±4.98a | 46.34±4.35abc | 116.85±6.13a | 343.46±22.74a |
| W1M0F1 | 19.87±5.24a | 5.19±0.65a | 94.11±15.81a | 81.92±8.76a | 48.11±3.08ab | 112.32±12.05a | 361.52±23.06a |
| W1M1F0 | 3.19±0.56b | 1.20±0.31b | 22.76±0.90b | 25.99±2.70b | 41.13±2.52bc | 111.17±8.09a | 205.44±2.78c |
| W1M1F1 | 20.69±3.57a | 5.65±0.81a | 83.91±3.61a | 79.95±5.52a | 54.36±5.84a | 110.18±14.65a | 354.74±21.09a |

The six treatments were: (1) the control treatment, without straw retention and mineral fertilizer (W0M0F0), (2) single application of mineral fertilizer (W0M0F1), (3) maize straw retention combined with mineral fertilization (W0M1F1), (4) wheat straw retention combined with mineral fertilizer (W1M0F1), (5) both wheat and maize straw retention with no fertilizer (W1M1F0), and (6) both wheat and maize straw retention combined with mineral fertilizer (W1M1F1) respectively. Inorganic P fractions includes calcium-bound P (Ca-P), aluminum-bound P (Al-P), iron-bound P (Fe-P), and occluded phosphate (O-P), Ca-P can be divided into dicalcium phosphate (Ca$_2$-P), octacalcium phosphate (Ca$_8$-P) and apatite (Ca$_{10}$-P). Values in each column followed by the different lowercase letters indicate significant differences (P < 0.05).





Table 3 Phosphorus K-edge XANES fitting results (%) showing the relative percent of each P species in water-extractable colloids (WECs)

| Samples | DCP | Al-P | Fe-P | IHP |
|---|---|---|---|---|
| W0M0F0 | 29.25±2.36a | 20.46±0.93b | 23.69±2.51b | 26.60±1.09a |
| W0M0F1 | 7.31±0.93b | 31.35±0.53a | 44.55±1.42a | 16.79±0.49b |
| W1M1F0 | 23.91±4.14a | 20.14±1.98b | 28.58±2.28b | 27.37±0.70a |

The six treatments were: (1) the control treatment, without straw retention and mineral fertilizer (W0M0F0), (2) single application of mineral fertilizer (W0M0F1), (3) maize straw retention combined with mineral fertilization (W0M1F1), (4) wheat straw retention combined with mineral fertilizer (W1M0F1), (5) both wheat and maize straw retention with no fertilizer (W1M1F0), and (6) both wheat and maize straw retention combined with mineral fertilizer (W1M1F1) respectively. DCP, dibasic calcium phosphate dihydrate (DCP, CaHPO$_4$·2H$_2$O); Al-P, aluminum phosphate (AlPO$_4$); Fe-P, iron phosphate dihydrate (FePO$_4$·2H$_2$O); and IHP, inositol hexakisphosphate, Values in each column followed by the different lowercase letters indicate significant differences (P < 0.05).



Table 4 Concentrations (mg/kg) of P species in bulk soil evaluated in the solution $^{31}$P NMR analysis

| Samples | NaOH-Na$_2$EDTA extracted P | Inorganic P | | Organic P | | | | | |
|---|---|---|---|---|---|---|---|---|---|
| | | Orth | Pyro | Orthophosphate monoesters | | | | Orthophosphate diesters | |
| | | | | Monoesters | Myo-IHP | Scyllo-IHP | Other mono | Diesters | Glyc+nucl |
| W0M0F0 | 120.47±11.00b | 62.26±0.23c | 5.60±0.02a | 41.40±1.17b | 7.16±0.47a | 1.56±0.45a | 32.68±2.08a | 11.21±0.92b | 10.59±0.92a |
| W0M0F1 | 309.62±30.41a | 221.21±4.47ab | 7.73±1.41a | 61.94±1.25ab | 13.27±0.27a | 4.42±0.09a | 44.24±0.89a | 18.76±4.31ab | 16.57±1.23a |
| W0M1F1 | 320.30±32.89a | 225.11±12.29ab | 5.67±1.90a | 68.27±10.58a | 11.26±0.61a | 4.50±0.25a | 52.51±11.44a | 21.26±3.61a | 19.09±0.55a |
| W1M0F1 | 340.18±40.35a | 244.85±7.47a | 7.35±0.22a | 68.40±8.30a | 12.14±6.55a | 3.70±1.84a | 52.56±3.59a | 19.59±0.60ab | 18.39±2.29a |
| W1M1F0 | 126.11±14.31b | 60.78±0.62c | 6.39±1.35a | 44.67±0.83b | 7.90±0.08a | 2.43±0.02a | 34.33±0.94a | 14.28±1.14ab | 11.54±0.74a |
| W1M1F1 | 286.84±29.14a | 208.68±5.37b | 5.20±1.34a | 54.18±4.51ab | 9.41±1.72a | 4.17±0.11a | 40.6±6.33a | 18.78±0.48ab | 17.72±1.02a |

The six treatments were: (1) the control treatment, without straw retention and mineral fertilizer (W0M0F0), (2) single application of mineral fertilizer (W0M0F1), (3) maize straw retention combined with mineral fertilization (W0M1F1), (4) wheat straw retention combined with mineral fertilizer (W1M0F1), (5) both wheat and maize straw retention with no fertilizer (W1M1F0), and (6) both wheat and maize straw retention combined with mineral fertilizer (W1M1F1) respectively. Calculation by including diester degradation products (i.e. Glyc+nucl: α/β- glycerophosphate, and mononucleotides) with orthophosphate diesters (Diesters) rather than orthophosphate monoesters (Monoesters). Phosphorus compounds include orthophosphate (Orth), pyrophosphate (Pyro), myo inositol hexakisphosphate (Myo-IHP), scylloinositol hexakisphosphate (Scyllo-IHP), other monoesters not specifically identified (Other mono), α/β- glycer-ophosphate (Glyc), and mononucleotides (nucl). Values in each column followed by the different lowercase letters indicate significant differences (P < 0.05).



Table 5 Concentrations (mg/kg) of P species in water-extractable colloids (WECs) evaluated in the solution $^{31}$P NMR analysis

| Samples | NaOH-Na$_2$EDTA extracted P | Inorganic P | | Organic P | | | | | | | |
|---|---|---|---|---|---|---|---|---|---|---|---|
| | | Orth | Pyro | Orthophosphate monoesters | | | | Orthophosphate diesters | | | |
| | | | | Monoesters | Myo-IHP | Scyllo-IHP | Other mono | Diesters | Glyc+nucl | DNA | |
| W0M0F0 | 258.36±19.99b | 96.97±12.00b | 14.02±1.05a | 110.24±6.77b | 17.28±0.58a | 4.32±0.15a | 88.63±6.04b | 37.14±6.29a | 28.58±4.63a | 0.97±0.12b | |
| W0M0F1 | 777.38±76.78a | 545.53±2.71a | 21.82±0.11a | 158.19±6.93a | 13.63±3.79a | 5.46±0.03a | 139.10±3.17a | 51.84±4.11a | 30.01±4.01a | 5.46±0.03a | |
| W1M1F0 | 280.02±28.65b | 111.96±9.46b | 16.40±5.33a | 110.56±10.38b | 17.78±1.65a | 4.48±0.38a | 88.31±9.10b | 41.09±4.42a | 29.96±3.78a | 1.12±0.09b | |

The six treatments were: (1) the control treatment, without straw retention and mineral fertilizer (W0M0F0), (2) single application of mineral fertilizer (W0M0F1), (3) maize straw retention combined with mineral fertilization (W0M1F1), (4) wheat straw retention combined with mineral fertilizer (W1M0F1), (5) both wheat and maize straw retention with no fertilizer (W1M1F0), and (6) both wheat and maize straw retention combined with mineral fertilizer (W1M1F1) respectively. Calculation by including diester degradation products (i.e. Glyc+nucl: α/β- glycerophosphate, and mononucleotides) with orthophosphate diesters (Diesters) rather than orthophosphate monoesters (Monoesters). Phosphorus compounds include orthophosphate (Orth), pyrophosphate (Pyro), myo inositol hexakisphosphate (Myo-IHP), scylloinositol hexakisphosphate (Scyllo-IHP), other monoesters not specifically identified (Other mono), α/β- glycer-ophosphate (Glyc), and mononucleotides (nucl). Values in each column followed by the different lowercase letters indicate significant differences (P < 0.05).