# Peer review of "Effect of straw retention and mineral fertilization on P speciation and P-transformation microorganisms in water extractable colloids of a Vertisol"

_EGUsphere, 2024_

## Author Comment (AC1)

**Response to Reviewer Comments**

**Responses to Reviewer 1:**

Comments to the Author

In this study, a fixed-site field trial was carried out from 2008 to 2021 to examine the impacts of straw (wheat and maize) retention and mineral fertilization (N, P, and K fertilizers) on soil inorganic P fractions, organic P species and P-transformation microorganisms in bulk soils and water-extractable colloid fractions. The paper presented a very exhaustive scientific work, The manuscript represents an important original contribution to research on soil phosphorus dynamics. As evaluation techniques, the methodology used and the results of excellent scientific quality. I recommend being accepted for publication after the minor revision. Please see the specific comments.

*Response: We thank the reviewers for the valuable comments which helped a lot for the manuscript improvement. We have addressed all of the suggestions and comments raised by the reviewers in the revised version. Please find below our detailed response to the comments.*

**Comment # 1:**

L141–Please report if some residue was left after digestion.

*Response: Thanks for your suggestion. No residue was left after digestion. We have revised the text accordingly.*
*Line 148-150: After microwave digestion, total P concentrations (TP) were determined by inductively coupled plasma optical emission spectroscopy (ICP-OES), with no residue left after digestion.*

**Comment # 2:**

L183–the spectra was not shown in Fig. S4?

*Response: OK, the spectra were shown in Fig. S6, not in Fig. S4. We have deleted it and the Fig. S6 was cited in the Line 271.*
*Line 194-195: The spectra of bulk soil and WECs were processed using MestReNova 10.0.2 software.*

**Comment # 3:**

Line 193–There is a misunderstanding in the description. Should be MEGAHIT was used to the assemble genome from reads (fastq formats)

*Response: OK, we have revised the description as suggested.*
*Line 204-205: MEGAHIT was used to assemble genome from the filtered reads (fastq formats) by de Bruijn graph with the minimum k-mer size of 21 (Li et al., 2015).*

**Comment # 4:**

Line 208–the normality distribution (Shapiro–Wilks test), not tests?

*Response: OK, we have corrected it to "Shapiro–Wilks test".*
*Line 219: The normality distribution (Shapiro–Wilks test) were performed before ANOVA.*

**Comment # 5:**

Fig.2– the caption needs to be corrected, What is "the A and B" in these figures?

*Response: Thank you for pointing that out. We have revised the caption of Fig. 2 accordingly.*
*Fig. 2 Relative abundance of genes responsible for microbial inorganic P solubilization, organic*

*P-mineralization, P-starvation regulation, and P-uptake and transport (A) and the individual gene relative abundance (B) in bulk soil*

**Comment # 6:**

Fig.4–What is "the A and B" in these figures?

*Response: Thank you for pointing that out. We have revised the caption of Fig. 4 accordingly.*

*Fig. 4 Relative abundance of genes responsible for microbial inorganic P solubilization, organic P-mineralization, P-starvation regulation, and P-uptake and transport (A) and the individual gene relative abundance (B) in bulk soils and water-extractable colloids (WECs) among the W0M0F0, W0M0F1, and W1M1F0 treatments.*

**Comment # 7:**

Table 3– modify the notes, there are three treatments in the tables.

*Response: OK. We have modified the notes in Table 3 as suggested.*

*The three treatments were: (1) the control treatment, without straw retention and mineral fertilizer (W0M0F0), (2) single application of mineral fertilizer (W0M0F1), and (3) both wheat and maize straw retention with no fertilizer (W1M1F0), respectively. DCP, dibasic calcium phosphate dihydrate (DCP, $CaHPO_4 \cdot 2H_2O$); Al-P, aluminum phosphate ($AlPO_4$); Fe-P, iron phosphate dihydrate ($FePO_4 \cdot 2H_2O$); and IHP, inositol hexakisphosphate, Values in each column followed by the different lowercase letters indicate significant differences ($P < 0.05$).*

**Comment # 8:**

Table 5–there are three treatments in the tables.

*Response: OK. We have modified the notes in Table 5 as suggested.*

*The three treatments were: (1) the control treatment, without straw retention and mineral fertilizer (W0M0F0), (2) single application of mineral fertilizer (W0M0F1), and (3) both wheat and maize straw retention with no fertilizer (W1M1F0) respectively. Calculation by including diester degradation products (i.e. Glyc+nucl: α/β-glycerophosphate, and mononucleotides) with orthophosphate diesters (Diesters) rather than orthophosphate monoesters (Monoesters). Phosphorus compounds include orthophosphate (Orth), pyrophosphate (Pyro), myo inositol hexakisphosphate (Myo-IHP), scylloinositol hexakisphosphate (Scyllo-IHP), other monoesters not specifically identified (Other mono), α/β- glycer-ophosphate (Glyc), and mononucleotides (nucl). Values in each column followed by the different lowercase letters indicate significant differences ($P < 0.05$).*

**Comment # 9:**

Table S2: glpA, glpB, glpC, glpK should be italic.

*Response: OK, we have italicized "glpA, glpB, glpC, glpK" in Table S2.*

**Comment # 10:**

Fig.S6– Solution $^{31}P$ NMR spectra of $NaOH$–$Na_2EDTA$ extracts of bulk soil (A) and water-extractable colloids (WECs, B), not bulk soil (a) and water-extractable colloids (WECs, b).

*Response: Thank you for your suggestion. We have revised the caption of Fig. S6.*

*Fig. S6 Solution $^{31}P$ NMR spectra of $NaOH$–$Na_2EDTA$ extracts of bulk soil (A) and water-extractable colloids (WECs, B)*

---

## Author Comment (AC2)

**Response to Reviewer Comments**

**Responses to Reviewer 2:**

Comments to the Author

This paper described the influences of straw retention and mineral fertilizer on the different phosphorous (P) forms in bulk soil and water-extractable colloid fractions, including the two P forms and their specific P specifies. The addition of P-related metagenomics analysis provides a comprehensive map of P cycling concerning different land management in the field. The results are very interesting to the current studies of P cycling in soils, with advanced technologies applied, which can help better understand P transformation in different fractions, but the paper could be more precise and would benefit from restructuring. The statistical methods you chose need to be further considered with respect to your observation size. The Result section is well written in general, but it would be better to modify some of the Figures and clarify some of your results. The major weakness of this paper, from my perspective, is the unclarification of your highlights, which means the Discussion section should be heavily revised. The bullet points in the Discussion section should be clearly delivered accordingly, a summarized paragraph is recommended for each subsection. Therefore, the recommendation for this manuscript is a major revision.

**Response: *Thanks for your constructive suggestions on our paper. We have revised the paper according to your suggestions. The following is the answers and revisions we have made in response to the questions and suggestions on an item-by-item basis. A detailed explanation of the revision follows below.***

**Comment # 1:**

Material and methods

Line 102- Once you decided to use the abbreviation of all your six treatments, use them consistently for the rest of the manuscript

***Response: Thanks for your suggestion. We used the abbreviation of six treatments consistently for the rest of the manuscript.***

**Comment # 2:**

Line 131: Are there reasons why you only fractionate the 3 treatments?

***Response: To further investigate the impact the sole straw retention (W1M1F0) and sole mineral fertilization (W0M0F1) on P cycling in soil colloids (WECs), the two treatments and the control treatment (W0M0F0) were fractionated.***

**Comment # 3:**

Line 132: Why do you use 'moist soil samples' for sedimentation? Do you also measure the soil texture?

***Response: Thanks for your suggestion.***

***We used the filed-fresh soil samples for sedimentation. We have added the description in Line 132-135: The field-fresh soil samples were used for sedimentation to replicate natural conditions where soil exists in its natural state, neither completely dry or saturated, enabling a more accurate study of these natural processes.***

***We also measured the soil texture, and added the description in Line 140-142:The soil was classified as sandy loam according to the international soil texture classification standard. The mass proportions of particles with >20 μm, 2-20 μm and <2 μm to bulk soil were shown in Fig. S1.***

**Comment # 4:**

Line 137: I think you mean 'The mass proportion of particles...'

*Response: Yes. We have revised it.*

*Line 141-142: The mass proportions of particles with >20 μm, 2-20 μm and <2 μm to bulk soil were shown in Fig. S1.*

**Comment # 5:**

Line 140: Which method and in which soil/solution ratio (w/v) do you use for pH measurement? As far as I know, there are big differences between the CaCl2, H2O2, and KCl methods. And which instrument do you use (specify all the instrument information you use for your analysis)?

*Response: We measured soil pH in a 1:2.5 soil/ ultrapure water suspension with Rex Electric Chemical PHSJ-3F. We have added it in the manuscript.*

*Line 144-145: A pH meter (Rex Electric Chemical PHSJ-3F) was utilized to measure soil pH in a 1:2.5 soil/ ultrapure water suspension.*

**Comment # 6:**

Line 140: Any pretreatment for SOC and TN measurements? The chemical measurements need to be briefly described (apply to the rest of the method section e.g. Line 145, Line 155-Line 156...)

*Response: We have revised it as follows:*

*Line 146-148: Prior to measuring SOC and TN, the samples were passed through a 0.149mm sieve. For SOC measurement, 1M HCl was added to the samples in small increments until effervescence stops (Schumacher, 2002).*

*Line 153-159: The extracted C with 0.5 M $K_2SO_4$ in non-fumigated and fumigated samples was determined with the Multi N/C 2100S TOC-TN analyzer. The dissolved organic carbon (DOC) was quantified as the extracted organic C by $K_2SO_4$ from the non-fumigated samples (Wu et al., 2019). MBC was quantified by measuring the variation in extractable C content between the non-fumigated and fumigated soil samples, using the universal conversion factor of 0.45. MBP was calculated as the variation in extractable P with 0.5 M $NaHCO_3$ between the non-fumigated and fumigated soil samples, with a conversion factor of 0.40.*

*Line 163-167: These fractions included $Ca_2$-P, extracted with 0.25 M $NaHCO_3$ (pH 8.0); $Ca_8$-P, extracted with 0.5 M $NH_4Ac$ (pH 4.2); Al-P, extracted with 0.5 M $NH_4F$ (pH 8.2); Fe-P, extracted with 0.1 M $NaOH-Na_2CO_3$ (pH 12.0); occluded-P (O-P), extracted with 0.3 M CD (sodium citrate-dithionite-sodium hydroxide, pH 13); and $Ca_{10}$-P, extracted with 0.25 M $H_2SO_4$ (pH 1.0).*

**Comment # 7:**

Line 169: Reconsider two 'considered'...

*Response: We deleted one 'considered ' in Line 179-180.*

*Line 179-180: The most likely P species was considered based on these results.*

**Comment # 8:**

Line 207: Specify the version of SPSS and R you use and cite the relevant references. For R, list the packages you use and find the relevant citations.

*Response: It is a constructive suggestion. We have provided the version of SPSS and R as well as the package.*

*Line 218-232: The IBM SPSS (version 25.0) and R (version 4.2.0) software were utilized for statistical analyses*

*and data visualization. The normality distribution (Shapiro–Wilks test) were performed before ANOVA. To identify significant differences among mean values at a significance level of 0.05, the Tukey's honestly significant differences (HSD) test was employed. The differences of soil properties, total P, inorganic P, organic P, ACP, and ALP between bulk soils and WECs were tested by independent-samples T test. The differences of P cycling genes composition in bulk soils and WECs were displayed by principal component analysis (PCA) with the R package "FactoMineR" (Lê Sébastien et al., 2008). Principal coordinate analysis (PCoA) was utilized to present the microbial bacterial β-diversity for typical P-solubilization (gcd) and mineralization (phoD) genes with the R package "vegan" and "ape" (Paradis and Schliep, 2019; Oksanen J, 2024). The associations between the abundances of P-transformation genes and soil characteristics were assessed using Spearman's correlations by R package "psych" with the correlation coefficients (R) > 0.6 and P-value <0.05 (Revelle, 2024). Structural equation modeling (SEM) was used to explore the relationships among agricultural managements, soil properties, and P-cycling related genes by Amos (24.0). The model fit was assessed with goodness of fit (GFI) and root square mean error of approximation (RMSEA).*

**Comment # 9:**

Line 216: SEM and PCA may not be applicable considering the size of your observation. Please at least check the degree of freedom to see if your interpretation, especially SEM results, can be regarded as reliable and stable. Otherwise, there are also similar models for a smaller observation size.

*Response: Thanks for your suggestion.*
*The degree of freedom (df) was 5.*
*The Chi-square/df was typically used to characterize good model fit. We have added it in Line 323-325:The Chi-square/df was 1.8 , which was less than 2 and indicated that the SEM model was a superior fit (Alavi et al., 2020).*

**Comment # 10:**

Results
Line 231: (Table 1) I guess
*Response: Yes. We have added Table 1.*
*Line 242-244: The application of sole straw retention (i.e., W1M1F0) had little effect on these soil properties except for slight increases in soil MBC and MBP contents compared with the control treatment (Table 1).*

**Comment # 11:**

Line 238: See above, either use the abbreviations or use the full name across the manuscript. And could you explain why you only measured the 3 treatments, not all of them? I also see differences from Table 1 between the 6 treatments and a significant increase in IP fractions, at least in W1M1F1, for example. Otherwise, I would think only include the 3 treatments in your results and discussion.
*Response: Thanks for your suggestion. The abbreviations were used consistently throughout the manuscript.*
*To further explore the impact the sole straw retention (W1M1F0) and sole mineral fertilization (W0M0F1) on P cycling in soil colloids (WECs), these two treatments and the control treatment (W0M0F0) of water extractable colloids were fractionated and investigated.*

*We analyzed the soil properties, P fractions and genes associated with P transformation in bulk soils across six treatments. For water-extractable colloids (WECs), these indicators were analyzed among three treatments (W0M0F1, W1M1F0 and W0M0F0). Focusing on these three representative treatments allow us to study the*

*specific mechanisms and effects of straw retention/mineral fertilization alone on phosphorus cycling at a microscopic colloidal scale.*

*Significant increases in soil properties and IP fractions were observed under mineral fertilization treatments (W0M0F1, W0M1F1, W1M0F1, W1M1F1), with no significant differences among these treatments. We then further investigated the impact of mineral fertilization alone (W0M0F1) on phosphorus cycling in soil colloids (WECs).*

**Comment # 12:**

Line 240: I personally would recommend avoiding overusing unclear words such as 'obvious', 'change', 'effect', 'little' etc., And better avoid 'for example' in the result section. You either present them or not.

*Response: We have revised it as suggested.*

*Line 251-254: The influence of either mineral fertilization or straw retention on physicochemical properties of WECs was more remarkable than their effects on bulk soils. Organic C and total N contents in WECs experienced a substantial rise following the implementation of straw retention compared to the control treatment, as depicted in Fig. 1 A and B.*

**Comment # 13:**

Line 283: 'This indicated that...' This, for me, looks like a Discussion, not a Result.

*Response: Thanks for your suggestion. The sentence has been deleted.*

**Comment # 14:**

Line 289: 'all the tested samples' make it clear, as you only fractionated 3 treatments, then better not use 'all'

*Response: OK, we have revised it as suggested.*

*Line298-299: The PCA analysis (Fig. 3 B) exhibited a clear segregation between the P-cycling genes in WECs and those in bulk soils for the W0M0F1, W1M1F0 and W0M0F0 treatments.*

**Comment # 15:**

Line 290: 'Staw retention'. which treatment do you mean? sole straw retention?

*Response: OK, we have revised it as suggested.*

*Line 299-301: Sole straw retention caused significant differences of relative abundance for many gene species including ppa, ppk, phoD, phoN, phy, phoR, phnCDE and ugpBAEC between WECs and bulk soils.*

**Comment # 16:**

Line 293: 'The control treatment caused significant...' This also happened in the previous description. The control should perform as a reference, which means it is not a treatment but a benchmark you need to compare firstly with other treatments (e.g. mineral fertilizer, sloe straw retention). The comparisons between other treatments (other than control) can be performed afterwards.

*Response: Thanks for your suggestion. We have deleted them.*

**Comment # 17:**

Line 299- Check grammer

*Response: We have checked grammar and revised it.*

*Line 306-310: The phoD gene (encoding alkaline phosphatases) and gcd gene (encoding glucose dehydrogenase*

*for synthesizing) serve as critical indicators of P mineralization and solubilization, respectively. As shown in Fig. 4, sole straw retention significantly increased the abundance of the phoD gene, whereas mineral fertilization significantly decreased the abundance of the gcd gene in WECs compared with bulk soils.*

**Comment # 18:**

Line 310- Does this mean that the other fractions are more sensitive to P cycling genes ? Is it just because of the low proportion of WECs?

*Response: The more P gene species were correlated with soil properties and nutrients in bulk soils than WECs, then the P genes species were more likely to be affected by soil properties and nutrient content in bulk soils.*
*This suggested that the response of P cycling genes to soil properties in bulk soil were more sensitive than those in WECs.*

**Comment # 19:**

Fig. 1: This is quite interesting that straw retention does not increase the OM in total , but only changes the partitioning of OM in different fractions. While, the sole mineral fertilizer significantly increases the OM in total. How do you harvest the plants with sole mineral fertilizer treatment?   Do you leave the roots or bottom stems in the field? Missing in the Method

*Response: For the sole mineral fertilizer treatment, straws were removed and the roots were left in the field. This information has been added in Line 112-113 of Materials and methods section.*
*Line 112-113: For the W0M0F1 treatments, straws were removed and the roots were left in the field.*

**Comment # 20:**

Fig. 2: Increase the text size of the plot and modify the note.
*Response: OK. We have increased the text size and modified the note in Fig.2.*

[Figure]

Fig. 2 Relative abundance of genes responsible for microbial inorganic P solubilization, organic P-mineralization, P-starvation regulation, and P-uptake and transport (A) and the individual gene relative abundance (B) in bulk soil

The relative abundances of genes were calculated related to the annotated reads. Significant differences between treatments in bulk soil are indicated by lowercase letters (p<0.05). The relative abundance of glp transporter systems was calculated as the average abundances of gene glpA, glpB, glpC, and glpK; the phn transporter systems was calculated as the average abundances of gene phnC, phnD, and phnE; the pst transporter systems was calculated as the average abundances of gene pstS, pstC, pstA, and pstB; The ugp transporter systems was calculated as the average abundances of gene ugpB, ugpA, ugpE, and ugpC.

**Comment # 21:**

Fig. 4: It is confusing. You tried to compare the two fractions, but it is hard to read the information and make the reader easily think you were comparing the six treatments... and lack of description of A and B. I suggest remaking the figure

*Response: OK. We have remade the Fig.4 and added the description of A and B.*

[Figure]

Fig. 4 Relative abundance of genes responsible for microbial inorganic P solubilization, organic P-mineralization, P-starvation regulation, and P-uptake and transport (A) and the individual gene relative abundance (B) in bulk soil and water-extractable colloids (WECs) among the W0M0F0, W0M0F1, and W1M1F0 treatments

The relative abundances of genes were calculated related to the annotated reads. Significant differences between treatments in bulk soil are indicated by lowercase letters (p<0.05). Significant differences between treatments in WECs (< 2μm) are indicated by capital letters (p<0.05). Significant differences between bulk soil and WECs are as follows, * p < 0.05 and ** p < 0.01 (Independent-samples T test). The relative abundance of glp transporter systems was calculated as the average abundances of gene glpA, glpB, glpC, and glpK; the phn transporter systems was calculated as the average abundances of gene phnC, phnD, and phnE; the pst transporter systems was calculated as the average abundances of gene pstS, pstC, pstA, and pstB; The ugp transporter systems was calculated as the average abundances of gene ugpB, ugpA, ugpE, and ugpC.

**Comment # 22:**

Fig. 5: See comments above... Again, you should always compare them with the control, though you want to compare between two fractions...

*Response: Thanks for your suggestion.*

*The phoD gene (encoding alkaline phosphatases) and gcd gene (encoding glucose dehydrogenase for synthesizing) serve as critical indicators of P mineralization and solubilization, respectively. As shown in Fig. 4, sole straw retention significantly increased the abundance of the phoD gene, whereas mineral fertilization significantly decreased the abundance of the gcd gene in WECs compared with bulk soils. Thus, we further performed the taxonomic assignments of phoD genes for the W1M1F0 treatment and gcd genes for the W0M0F1 treatment between bulk soils and WECs.*

*The fig.5 highlighted the differences in taxonomic assignments of gcd gene for the W1M1F0 treatment and phoD gene for the W0M0F1 treatment between bulk soils and WECs.*

[Figure]

Fig. 5 Taxonomic assignments at the phylum level of the *phoD* gene for the W1M1F0 treatment (A), and the *gcd* gene for the W0M0F1 treatment (B) in bulk soil and water-extractable colloids (WECs)

**Comment # 23:**

Fig. 6 See comments above... check degree of freedom

*Response: The degree of freedom (df) was 5.*

*The Chi-square/df was typically used to characterize good model fit. We have added it in Line 323-325:The Chi-square/df was 1.8 , which was less than 2 and indicated that the SEM model was a superior fit (Alavi et al., 2020).*

**Comment # 24:**

Table 3: There are only 3 treatments in the table

*Response: OK, we have modified the Table 3 as suggested.*

Table 3 Phosphorus K-edge XANES fitting results (%) showing the relative percent of each P species in water-extractable colloids (WECs) among the W0M0F1, W1M1F0 and W0M0F0 treatments

| Samples | DCP | Al-P | Fe-P | IHP |
|---|---|---|---|---|
| W0M0F0 | 29.25±2.36a | 20.46±0.93b | 23.69±2.51b | 26.60±1.09a |
| W0M0F1 | 7.31±0.93b | 31.35±0.53a | 44.55±1.42a | 16.79±0.49b |
| W1M1F0 | 23.91±4.14a | 20.14±1.98b | 28.58±2.28b | 27.37±0.70a |

The three treatments were: (1) the control treatment, without straw retention and mineral fertilizer (W0M0F0), (2) single application of mineral fertilizer (W0M0F1), and (3) both wheat and maize straw retention with no fertilizer (W1M1F0), respectively. DCP, dibasic calcium phosphate dihydrate (DCP, $CaHPO_4 \cdot 2H_2O$); Al-P, aluminum phosphate ($AlPO_4$); Fe-P, iron phosphate dihydrate ($FePO_4 \cdot 2H_2O$); and IHP, inositol hexakisphosphate, Values in each column followed by the different lowercase letters indicate significant differences (P < 0.05).

**Comment # 25:**

Table 5: There are only 3 treatments in the table
*Response: OK, we have modified the Table 5 as suggested.*

Table 5 Concentrations (mg/kg) of P species in water-extractable colloids (WECs) evaluated in the solution $^{31}$P NMR analysis among the W0M0F1, W1M1F0 and W0M0F0 treatments

| Samples | NaOH-Na₂EDTA extracted P | Inorganic P | | Organic P | | | | | | |
|---|---|---|---|---|---|---|---|---|---|---|
| | | | | Orthophosphate monoesters | | | | Orthophosphate diesters | | |
| | | Orth | Pyro | Monoesters | Myo-IHP | Scyllo-IHP | Other mono | Diesters | Glyc+nucl | DNA |
| W0M0F0 | 258.36±19.99b | 96.97±12.00b | 14.02±1.05a | 110.24±6.77b | 17.28±0.58a | 4.32±0.15a | 88.63±6.04b | 37.14±6.29a | 28.58±4.63a | 0.97±0.12b |
| W0M0F1 | 777.38±76.78a | 545.53±2.71a | 21.82±0.11a | 158.19±6.93a | 13.63±3.79a | 5.46±0.03a | 139.10±3.17a | 51.84±4.11a | 30.01±4.01a | 5.46±0.03a |
| W1M1F0 | 280.02±28.65b | 111.96±9.46b | 16.40±5.33a | 110.56±10.38b | 17.78±1.65a | 4.48±0.38a | 88.31±9.10b | 41.09±4.42a | 29.96±3.78a | 1.12±0.09b |

The three treatments were: (1) the control treatment, without straw retention and mineral fertilizer (W0M0F0), (2) single application of mineral fertilizer (W0M0F1), and (3) both wheat and maize straw retention with no fertilizer (W1M1F0) respectively. Calculation by including diester degradation products (i.e. Glyc+nucl: α/β- glycerophosphate, and mononucleotides) with orthophosphate diesters (Diesters) rather than orthophosphate monoesters (Monoesters). Phosphorus compounds include orthophosphate (Orth), pyrophosphate (Pyro), myo inositol hexakisphosphate (Myo-IHP), scylloinositol hexakisphosphate (Scyllo-IHP), other monoesters not specifically identified (Other mono), α/β- glycer-ophosphate (Glyc), and mononucleotides (nucl). Values in each column followed by the different lowercase letters indicate significant differences (P < 0.05).

**Comment # 26:**

Discussion
The subtitles should be informative and clear, with bullet points emphasized. In addition, all the figures and tables described and shown in the Result section need to be used and properly referenced in the Discussion section; otherwise, the results do not need to be in the Result section. Cite only the necessary literature.
*Response: Thanks for your suggestion.*
*The subtitles were revised as suggested:*
**4.1 Mineral fertilization restricted genes involved in P transformation in bulk soils**
**4.2 Straw retention increased the abundances of *phoD* gene and *phoD*-harbouring *Proteobacteria* in WECs**

*The figures and tables were properly referenced in the Discussion section. The necessary literature was cited.*

**Comment # 27:**

Line 326- 'to the enhanced organic matter from crops...' see comments above, how did you harvest the crops with

sole mineral fertilizer input?

*Response: Thanks for your suggestion. For the sole mineral fertilizer treatment, straws were removed and the roots were left in the field.*

*We have added it in Line 112-113 of the Materials and methods section.*

*Line 112-113: For the W0M0F1 treatments, straws were removed and the roots were left in the field.*

**Comment # 28:**

Line 342: 'Consistent with our findings...' Normally, it should be 'consistent with previous studies, our study finds...'. The discussion section is to interpret your data and deliver your findings/opinions to the audience instead of describing other studies.

*Response: We have revised it as follows:*

*Line 350-353: Consistent with previous findings (Ikoyi et al., 2018; Dai et al., 2020), mineral fertilization alone or combined with straw retention reduced the abundance of genes about P mineralization (e.g., phoA, phoD, phy, ugpQ), P-starvation regulation (e.g., phoR), P-uptake and transport (e.g., phnCDE) significantly (Fig. 2).*

**Comment # 29:**

Line 344: 'Long-term P...' is this also mineral P fertilizer?

*Response: Yes. 'Long-term P...' is also mineral P fertilizer in the paper (Dai et al., 2020).*

**Comment # 30:**

Line 346-353: Where is your own data? Please reference it and demonstrate it.

*Response: OK, we have revised as suggested.*

*Line 354-364: Additionally, Chen et al. (2017) identified soil pH as the primary factor influencing the compositions of microbial community harboring the phoD gene, noting a positive correlation between soil pH and of the phoD gene abundance. Studies have provided evidence that a decrease in soil pH could inhibit bacterial/fungal growth (Li et al., 2020), modify the microbial community compositions (Rousk et al., 2010), and decrease the relative abundances of Actinobacteria and Proteobacteria for phoD gene (Luo et al., 2017), which in turn decreases P mineralization capacity. In this study, Spearman's Rank correlations showed the phoD, phoA, phy, ugpQ, and phoR genes abundances were correlated negatively with the contents of orthophosphate, orthophosphate monoesters, orthophosphate diesters, and positively with soil pH ($p < 0.05$) (Fig. S8 A). Thus, the decline in the abundance of P-cycling related genes (Fig. 2) can be attributed to increased soil P contents and low soil pH (Table 1 and 4) under mineral fertilization compared with the control treatment.*

**Comment # 31:**

Line 359: I assume you tried to compare with the control treatment?

*Response: Yes. We have added it.*

*Line 362-364: Thus, the decline in the abundance of P-cycling related genes (Fig. 2) can be attributed to increased soil P contents and low soil pH (Table 1 and 4) under mineral fertilization compared with the control treatment.*

**Comment # 32:**

Line 365: If you want to discuss this, then you need to describe it first the CN or CP ratio first in the Result section and reference it in the Discussion section accordingly.

*Response: Thanks for your suggestion.*

*The C/N in wheat and maize straw was not measured and it was in the range of 52-73:1 according the previous studies. This information was provided in Line 369-371: The C/N in wheat and maize straw (52-73:1) were significantly higher than suitable microorganisms C:N (25-30:1) for straw decomposition (Cai et al., 2018), and microorganisms needed to consume soil original N when decomposing straw.*

**Comment # 33:**

Line 368: Conclued from Table 1?

*Response: Yes. We have added it.*

*Line 372-373:Thus, the straw retention for 13 years did not show any significant impact on soil C, N, P nutrients (Table 1).*

**Comment # 34:**

Line 371: 'The slight increase ...' confused

*Response: Thanks for your suggestions. We have revised it.*

*Line 377-378: The increase in MBC resulted in the increase of MBP (Spohn and Kuzyakov, 2013), as shown in Table 1.*

**Comment # 35:**

Line 372: Combine your own data

*Response: Thanks for your suggestion. We have added 'as shown in Table 1' in Line 377-378.*

*Line 377-378: The increase in MBC resulted in the increase of MBP (Spohn and Kuzyakov, 2013), as shown in Table 1.*

**Comment # 36:**

Line 376: 'Had remarkable influences on...' make it clear, the influence is increase or decrease?

*Response: Thanks for your suggestion. We have revised it as suggested.*

*Line 381-383: In this study, straw retention incorporated with mineral fertilization brought remarkable decreases in soil pH and significant increases in soil nutrients, which was significantly different from sole straw retention.*

**Comment # 37:**

Line 378-379: It is better to have a solid summary based on your data and findings. This finding is not new as the OM decomposition will release organic acid. I expect to see more interesting results from your data.

*Response: We have revised it as suggested.*

*Line 384-386: Sole straw retention showed minimal effects on soil properties, P species and transformation genes in bulk soil. Interestingly, it has started to have a notable influence on these indicators in the soil colloids (WECs), as discussed below.*

**Comment # 38:**

Line 381: 'AP'?

*Response: AP is the available P. The abbreviation of AP was described in Line 150-151: Available P (AP, Olsen-P) concentration was quantified by Olsen and Sommers (1982).*

**Comment # 39:**

Line 383: 'The influences of...' Make your opinions clear with referenced results; how do you define the stronger effect?

*Response: We have revised the description as suggested.*

*Line 389-396: The higher concentrations of SOC, TN, TP, AP and various P species in WECs (Fig. 1 and Table 5) compared with bulk soil (Table1 and 4) indicated that nutrients are enriched within the WECs due to their high specific surface area (Jiang et al., 2014). Mineral fertilization and straw retention caused significant increases in these indicators within the WECs compared to bulk soil, suggesting that the managements practices exerted more significant impacts on soil properties and P species within the WECs when compared to the effects observed in bulk soils. This highlighted the heightened sensitivity of the physicochemical properties of soil microparticles to environmental disturbances compared to bulk soil.*

**Comment # 40:**

Line 394: 'slight increase' is it significant?

*Response: The slight increase meant the concentrations of TP and each P species for WECs increased, while there was no statistics significant difference between WECs and bulk soil.*

**Comment # 41:**

Line 395: 'considerable influence...' How much is the influence? increase or decrease?

*Response: Thank you for pointing that out. We have revised it as suggested.*

*Line 404-406: This indicated that straw retention promoted the accumulation of nutrients on WECs, which could enhance the supply and cycling of P.*

**Comment # 42:**

Line 399: 'Research conducted...' It seems these are not related to your context.

*Response: Thank you for pointing that out. We have modified the citation and revised it as suggested.*

*Line 407-420:Straw retention caused the greater change of P cycling genes between WECs and bulk soils compared with mineral fertilization (Fig. 4 B) and led to a significant increase of phoD gene in WECs compared with bulk soils. For bacterial taxa containing phoD gene, the abundance of Proteobacteria (Fig. 5 A) increased significantly in WECs compared with those in bulk soils under sole straw retention. This indicated that straw retention might increase the phoD gene abundance by influencing phoD-harbouring Proteobacteria, and then increase P mineralizing capacity in WECs. Several studies have highlighted that Proteobacteria has been recognized as a crucial group of microorganisms involved in the mineralization of P (Zhang et al., 2023) and the increase in phoD-harbouring Proteobacteria could improve potential P mineralization (Xie et al., 2020). The Proteobacteria belongs to copiotrophic microorganisms groups, and accumulates in rich nutrient soils (Wang et al., 2022). Research conducted by Fierer et al. (2012) and Ling et al. (2014) have shown that higher concentrations of total N, P and organic C could promote the growth of such microorganisms. In our research, the notable increases in SOC, TN and each P specie in WECs under straw retention likely created favorable conditions for the proliferation of copiotrophic bacteria (e.g., Proteobacteria).*

**Comment # 43:**

Line 410: 'clay particles'? <2 micrometers will be enough, in my opinion.

*Response: We have revised it as suggested.*

*Line 420-421: Generally, the WECs (clay particles) including natural organic matter (e.g., humus) and*

*inorganic colloids (silicate, and Al/Fe oxides)*

**Comment # 44:**

Line 419: Check the logic framework and how you get to this conclusion.

*Response: OK. We have revised it as suggested.*

*Line 407-430:Straw retention caused the greater change of P cycling genes between WECs and bulk soils compared with mineral fertilization (Fig. 4 B) and led to a significant increase of phoD gene in WECs compared with bulk soils. For bacterial taxa containing phoD gene, the abundance of Proteobacteria (Fig. 5 A) increased significantly in WECs compared with those in bulk soils under sole straw retention. This indicated that straw retention might increase the phoD gene abundance by influencing phoD-harbouring Proteobacteria, and then increase P mineralizing capacity in WECs. Several studies have highlighted that Proteobacteria has been recognized as a crucial group of microorganisms involved in the mineralization of P (Zhang et al., 2023) and the increase in phoD-harbouring Proteobacteria could improve potential P mineralization (Xie et al., 2020). The Proteobacteria belongs to copiotrophic microorganisms groups, and accumulates in rich nutrient soils (Wang et al., 2022). Research conducted by Fierer et al. (2012) and Ling et al. (2014) have shown that higher concentrations of total N, P and organic C could promote the growth of such microorganisms. In our research, the notable increases in SOC, TN and each P specie in WECs under straw retention likely created favorable conditions for the proliferation of copiotrophic bacteria (e.g., Proteobacteria). Generally, the WECs including natural organic matter (e.g., humus) and inorganic colloids (silicate and Al/Fe oxides) (Zhang et al., 2021) were considered to be the best natural microorganism adsorbents (Zhao et al., 2014; Madumathi, 2017). Previously conducted research has indicated that most bacteria (65%) associated with <2 μm soil particulates (Oliver et al., 2007). The population of the bacteria (Pseudomonas putida) attached to the clay particle in Red soil (Ultisol) was significantly higher compared to the populations found on silt and sand particles (Wu et al., 2012). Furthermore, the increased SOC could improve the surface area and activity of WECs (Zhao et al., 2014), thus increasing microorganism adhesion (Van Gestel et al., 1996). SOC was a key component of P binding in colloids (Sun et al., 2023). Thus, we considered that the P cycling microorganisms in soil colloids might be influenced by itself characteristics and the increased the nutrients contents of WECS under straw retention.*

**Comment # 45:**

Line 422: Check the logic framework and how you get to this conclusion.

*Response: We have revised it as suggested.*

*Line 431-436: In this study, mineral fertilization also caused the enhancements of SOC contents in WECs (Fig. 1), which positively influenced the abundance of P cycling genes. However, it was also noted that mineral fertilization led to a dramatic increase in P contents and a substantial decrease in soil pH by 1.76-1.89 units (Table 1), which restricted the expression and activity of P cycling genes in both WECs and bulk soils, as discussed before. Therefore, the difference of P-cycling genes between WECs and bulk soil under mineral fertilization was less significant than those under straw retention.*

**References**

Alavi M, Visentin D C, Thapa D K, Hunt G E, Watson R, Cleary M. 2020. Chi-square for model fit in confirmatory factor analysis. *Journal of Advanced Nursing*. **76**: 2209-2211.

Chen X D, Jiang N, Chen Z H, Tian J H, Sun N, Xu M G, Chen L J. 2017. Response of soil *phod* phosphatase gene to long-term combined applications of chemical fertilizers and organic materials. *Applied Soil Ecology*. **119**:

197-204.

Dai Z M, Liu G F, Chen H H, Chen C R, Wang J K, Ai S Y, Wei D, Li D M, Ma B, Tang C X, Brookes P C, Xu J M. 2020. Long-term nutrient inputs shift soil microbial functional profiles of phosphorus cycling in diverse agroecosystems. *The ISME Journal.* **14**: 757-770.

Ikoyi I, Fowler A, Schmalenberger A. 2018. One-time phosphate fertilizer application to grassland columns modifies the soil microbiota and limits its role in ecosystem services. *Science of The Total Environment.* **630**: 849-858.

Jiang C L, Séquaris J-M, Wacha A, Bóta A, Vereecken H, Klumpp E. 2014. Effect of metal oxide on surface area and pore size of water-dispersible colloids from three german silt loam topsoils. *Geoderma.* **235-236**: 260-270.

Li C Y, Hao Y h, Xue Y L, Wang Y, Dang T H. 2020. Effects of long-term fertilization on soil microbial biomass carbon, nitrogen, and phosphorus in the farmland of the loess plateau, china. *Journal of Agro-Environment Science.* **39**: 1783-1791.

Luo G, Ling N, Nannipieri P, Chen H, Raza W, Wang M, Guo S, Shen Q. 2017. Long-term fertilisation regimes affect the composition of the alkaline phosphomonoesterase encoding microbial community of a vertisol and its derivative soil fractions. *Biol Fertil Soils.* **53**: 375-388.

Rousk J, Bååth E, Brookes P C, Lauber C L, Lozupone C, Caporaso J G, Knight R, Fierer N. 2010. Soil bacterial and fungal communities across a ph gradient in an arable soil. *The ISME Journal.* **4**: 1340-1351.

Schumacher B. 2002. Methods for the determination of total organic carbon (toc) in soils and sediments. *Ecological Risk Assessment Support Center Office of Research and Development.*

Wu L, Zhang W J, Wei W J, He Z L, Kuzyakov Y, Bol R, Hu R G. 2019. Soil organic matter priming and carbon balance after straw addition is regulated by long-term fertilization. *Soil Biology and Biochemistry.* **135**: 383-391.

---

## Author Response (AR2)

**Cover letter**

**Dear editors,**

Thank you for your letter and the constructive suggestions from the reviewers regarding our manuscript. These comments are invaluable and will greatly assist us in revising and improving our paper. In light of the suggestions and the reviewers' comments, we have revised the paper entitled *"Effect of straw retention and mineral fertilization on P speciation and P-transformation microorganisms in water extractable colloids of a Vertisol" (Manuscript number: EGUSPHERE-2024-983)*. The revised portion are marked in track change in the manuscript. Appended to this letter is our point-by-point response to the comments raised by the reviewers. If you have any questions regarding this paper, please do not hesitate to contact us.

We also would like to thank you for allowing us to resubmit a revised copy of the manuscript and hope that the revised manuscript will be acceptable for publication in *Biogeosciences*.
Looking forward to hearing from you as soon as possible.

Your Sincerely,
Xiaoqian Jiang

**Response to Reviewer Comments**

**Responses to Reviewer 1:**

Comments to the Author

This revised manuscript has significantly improved, but many minor construction mistakes and grammar issues need to be carefully and thoroughly revised. Therefore, the recommendation for this manuscript at this stage is a minor revision.

*Response: We thank the reviewers for their valuable comments, which have greatly contributed to the improvement of the manuscript. We have addressed all suggestions and comments raised by the reviewers in the revised version. Below, please find our detailed responses to the comments.*

**Comment # 1:**

Abstract

Line 26: increasing which P? Total P I guess?

*Response: Thank you for your suggestion. We have added "total P" in Line26.*

*Line 22-26:In bulk soil, mineral fertilization led to increases in the levels of total P, available P, acid phosphatase (ACP), high-activity inorganic P fractions (Ca2-P, Ca8-P, Al-P, and Fe-P) and organic P (orthophosphate monoesters and orthophosphate diesters), but significantly decreased the abundances of P cycling genes including P mineralization, P-starvation response regulation, P-uptake and transport by decreasing soil pH and increasing total P in bulk soil.*

**Comment # 2:**

Line 29: Could you please specify 'change'?

*Response: Thank you for your suggestion. We have revised it as suggested.*

*Line 28-30: Furthermore, straw retention caused significant differences of relative abundances for more P cycling genes between WECs and bulk soils than mineral fertilization.*

**Comment # 3:**

Line 31: It will be better to add one summary sentence. 'Thus, …' only contains the results from straw retention, lack of context regarding to the mineral fertilization as you mentioned in the title.

*Response: Thank you for pointing it out. We have added the summary sentence.*

*Line 31-34: Thus, mineral fertilization reduced microbial P-solubilizing and mineralizing capacity in bulk soil. Straw retention could potentially accelerate the turnover, mobility and availability of P by increasing the nutrient contents and P mineralizing capacity at the microscopic colloidal scale.*

**Comment # 4:**

Material and Methods

Line 146: If you decide to use abbreviations like SOC, TN here, you need to clarify them at the beginning of your manuscript and use the abbreviations constantly throughout the manuscript. In the abstract as well.

*Response: We have clarified the abbreviations at the beginning of the manuscript as follows:*

*SOC: Soil Organic Carbon*

*TN: Total Nitrogen*

*Additionally, we have ensured that these abbreviations are used consistently throughout the manuscript.*

**Comment # 5:**

Line 155: Abbreviation? Check throughout the whole manuscript.
*Response: Thank you for pointing that out. We have deleted the abbreviation (DOC) in Line 155.*

**Comment # 6:**

6 Discussion
The subtitles are informative. The construction of the discussion has been remarkably improved. But there are still many minor grammar and logical issues. Fig.2 and Fig. 4 need to be recognizable to readers, so I suggest greatly increasing the font size.
*Response: It is a good suggestion. We have increased the font size in Fig. 2 and Fig. 4 to enhance readability.*

[Figure]

Fig. 2 Relative abundance of genes responsible for microbial inorganic P solubilization, organic P-mineralization, P-starvation regulation, and P-uptake and transport (A) and the individual gene relative abundance (B) in bulk soil

[Figure]

Fig. 4 Relative abundance of genes responsible for microbial inorganic P solubilization, organic P-mineralization, P-starvation regulation, and P-uptake and transport (A) and the individual gene relative abundance (B) in bulk soil and water-extractable colloids (WECs) among the W0M0F0, W0M0F1, and W1M1F0 treatments

**Comment # 7:**

Line 333-334: For me, 'mineral fertilization decreased soil pH, increase soil TP, thus decreasing the …' has a logical issue. I think these are parallel results obtained from Table 1 and Fig. 2, the causal relationship needs correlation results like Fig. 6.

*Response: Thank you for pointing that out. We have added the Fig.6 in Line 335.*

*Line 334-335: In bulk soil, mineral fertilization decreased soil pH, increased soil TP, thus decreasing the abundances of P transformation genes (Fig. 6).*

**Comment # 8:**

Line 338: Not relevant to your core topic

*Response: OK. We have modified the description in Line 336-339.*

*Line 336-339: The significant increases in soil organic matter and nutrients concentrations under mineral fertilization might be closely associated with the enhanced organic matter from crop residues, root exudates and the input of fertilizers (Tong et al., 2019).*

**Comment # 9:**

Line 407: See comments above.

*Response: OK. We have revised it as suggested.*

*Line 407-409: Straw retention caused significant differences of relative abundances for more P cycling genes between WECs and bulk soils than mineral fertilization (Fig. 4 B) and led to a significant increase of phoD gene in WECs compared with bulk soils.*

**Comment # 10:**

Line 346-353: Where is your own data? Please reference it and demonstrate it.

*Response: Thank you for your suggestion. We have referenced our data in the manuscript and demonstrated it in Fig. 2.*

*Line 348-353: Therefore, in the control and straw retention treatments with lower P concentrations, higher abundances of phoD, phy, phoR, and ugpQ genes were observed in comparison with the mineral fertilization treatments (Fig. 2). Consistent with previous findings (Ikoyi et al., 2018; Dai et al., 2020), mineral fertilization alone or combined with straw retention reduced the abundance of genes about P mineralization (e.g., phoA, phoD, phy, ugpQ), P-starvation regulation (e.g., phoR), P-uptake and transport (e.g., phnCDE) significantly (Fig. 2).*

**Comment # 11:**

Conclusion

The conclusion section should contain the clear main results the same as the Abstract. The summarized paragraphs at the end of each Discussion part can be summarized here. Meanwhile, the conclusion should explain the contribution of your research to the knowledge gap you introduced at the start of the abstract and introduction, and some perspectives if there are any.

*Response: Thank you for your suggestion. We have revised the conclusion.*

*Line 454-464:* **This study provides systematic insights into P speciation and P transformation microorganisms at the soil microparticle scale (WECs) compared with bulk soil under straw retention and mineral fertilization. Mineral fertilization decreased soil pH, increased soil TP, thus restricting genes involved in P transformation in bulk soils. Straw retention caused more obvious impact on the accumulation of organic C and total N of WECs and the greater change of P cycling genes between WECs and bulk soils even than mineral fertilization. The significant increase in the abundance of gene encoding for alkaline phosphatase (*phoD*) and *phoD*-harbouring *Proteobacteria* for WECs compared with bulk soils indicated the improved P mineralization capacity of WECs under straw retention. This information provided strong evidences that straw retention could potentially affect the turnover, mobility and availability of P mainly by changing the physicochemical and biochemical processes involved in the P transformation of soil colloids.**

---

## Author Response (AR3)

**Cover letter**

**Dear editors,**

Thank you for your letter and the constructive suggestions from the reviewers regarding our manuscript. These comments are invaluable and will greatly assist us in revising and improving our paper. In light of the suggestions and the reviewers' comments, we have revised the paper entitled ***"Effect of straw retention and mineral fertilization on P speciation and P-transformation microorganisms in water extractable colloids of a Vertisol" (Manuscript number: EGUSPHERE-2024-983)***. The revised portion are marked in track change in the manuscript. Appended to this letter is our point-by-point response to the comments raised by the reviewers. If you have any questions regarding this paper, please do not hesitate to contact us.

We also would like to thank you for allowing us to resubmit a revised copy of the manuscript and hope that the revised manuscript will be acceptable for publication in ***Biogeosciences***.
Looking forward to hearing from you as soon as possible.

Your Sincerely,
Xiaoqian Jiang

**Response to Reviewer Comments**

**Responses to Reviewer 1:**

thank you for considering and following the reviewer's suggestions, in particular to improve the Figures. However, I would request some further changes, and as also already previously pointed out by the reviewers earlier, I would recommend to have some quick language check.

Moreover, please also add some more details in the methods sections (some details are specified below in the line-by-line section). Please pay particular attention especially on the section 2.6 DNA extraction and metagenomics analysis. The details provided are still quite vague in some sections and it would be important to add more technical details. Please mention how the reads were filtered, and provide details on the pre-processing steps and a brief statement with details on the Diamond settings that have been used.

Finally, I would highly encourage to restructure the conclusions sections. At the moment the mostly summarize the results, only the very last sentence is rather a conclusion. To make the manuscript stronger, I would recommend that the authors could rephrase and rather focus on impacts or implications of the outcomes of their study. Below are some points I would like you to resolve specifically.

*Response: Thank you very much for your careful review and valuable suggestions. We highly appreciate your efforts in helping us improve our manuscript.*

*Regarding your request for further changes to improve the figures, we have improved the figures and enhance them as per your instructions.*

*As for the language check, we have conducted a thorough review and correction to ensure the language is accurate and fluent.*

*In the methods section, we have added more details as you have requested.*

*We have rephrased the conclusions and focus on the impacts and implications of our study outcomes.*

*Conclusion:*

This study provides valuable insights into P speciation and the role of P transformation microorganisms at the soil microparticle scale (WECs) in the context of straw retention and mineral fertilization. Our findings underscore the critical influence of these management practices on soil chemistry and microbial dynamics. The decrease in soil pH and increases in soil TP under mineral fertilization hinder the expression of genes related to P transformation in bulk soils, potentially limiting the efficiency of P cycling. In contrast, straw retention enhances the accumulation of organic C and total N in soil colloids scale significantly, thus causing significant increase in the abundance of gene encoding for alkaline phosphatase (*phoD*) and *phoD*-harbouring *Proteobacteria* for WECs. It indicates that straw retention could potentially improve P availability by increasing P mineralization capacity of WECs. This information provided innovative evidence that straw retention could potentially affect the turnover, mobility and availability of P mainly by changing the physicochemical and biochemical processes involved in the P transformation of soil colloids.

**Comment # 1:**

Line 26: remove the last 'in bulk soil'.

*Response: Thank you for your suggestion. We have removed "in bulk soil" in Line26.*

**Comment # 2:**

Line 28: change to: ', but lead to increases in..'

*Response: Thank you for your suggestion. We have revised it as suggested.*

**Comment # 3:**

Line 29: this sentence is not really easy to understand, which treatment caused higher P cycling gene abundance compared to which?

*Response:*

*We appreciate your insights, and we have revised the sentence.*

*Line 29-31: Furthermore, compared with mineral fertilization, straw retention caused significantly greater differences in the relative abundances of P cycling genes between WECs and bulk soils..*

**Comment # 4:**

Line 47: when you use the Ca-P, Al-P etc... please first time spell them out or explain a bit better.

*Response: We have revised the manuscript to spell out and provide a brief explanation of the terms $Ca_2$-P, Al-P, Fe-P and $Ca_{10}$-P.*

*Line 46-49: Under mineral fertilization and straw retention, dicalcium phosphate ($Ca_2$-P), iron-bound P (Fe-P) and aluminum-bound P (Al-P) contents increased, but apatite ($Ca_{10}$-P) concentration reduced, thereby promoting the transformation of P fractions (Xu et al., 2022).*

**Comment # 5:**

Line 63: delete the space before 0.25

*Response: Thank you for pointing that out. We have deleted the space before 0.25.*

Comment # 6:

Line 91: this first sentence is a bit out of context here. Maybe change to:

In a long-term (13 year) field experiment modulating straw retention and mineral fertilization, we investigated 1) the responses....

*Response: We have revised the sentence as suggested.*

**Comment # 7:**

Line 95: This sentences could be removed, as they are almost repeating the same message as just before the research aim (Lines 89-90)

*Response: Thank you for pointing that out. We have removed the sentence.*

**Comment # 8:**

Line 124: Change to: From all six treatment plots soil samples were collected after wheat harvest in June 2021

*Response: OK. We have modified the description in Line 124.*

*Line 124: From all six treatment plots soil samples were collected after wheat harvest in June 2021.*

**Comment # 9:**

Line 128: delete 'the' before 'acid and alkaline phosphatase'

*Response: OK. We have delete 'the' in Line 128.*

*Line 126-128: The first subsample was preserved at 4 ∘C to examine soil microbial biomass C (MBC) and microbial biomass P (MBP), along with acid and alkaline phosphatase activities (ACP and ALP).*

**Comment # 10:**

Line 151: please add here, to remove inorganic carbon (and change stop to stopped).

*Response: Thank you for your suggestion. We have revised the sentence as suggested.*

*Line 149-150: For SOC measurement, 1M HCl was added to the samples in small increments until effervescence stopped to remove inorganic carbon (Schumacher, 2002).*

**Comment # 11:**

Line 154: change to 'using the method described by Olsen...'

*Response: Thank you for your suggestion. We have revised the sentence.*

*Line 152-153:* **Available P (AP, Olsen-P) concentration was quantified using the method described by Olsen and Sommers (1982).**

**Comment # 12:**

Line 155: leave the abbreviation in brackets: dissolved organic carbon (DOC).

*Response: Thank you for your feedback. We have added the abbreviation in brackets.*

*Line 156-157:* **The dissolved organic C (DOC) was quantified as the extracted organic C by K2SO4 from the non-fumigated samples (Wu et al., 2019).**

**Comment # 13:**

Line 160 and 162: provide references for the conversion factor

*Response: Thank you for pointing that out. We have provided references for the conversion factor.*

*Line 158-161:* **MBC was quantified by measuring the variation in extractable C content between the non-fumigated and fumigated soil samples, using the universal conversion factor of 0.45 (Vance et al., 1987). MBP was calculated as the variation in extractable P with 0.5 M NaHCO$_3$ between the non-fumigated and fumigated soil samples, with a conversion factor of 0.40 (Brookes et al., 1982).**

**Comment # 14:**

Line 200: change to: DNA was extracted using a Fast...

*Response: Thank you for pointing that out. We have revised the sentence.*

*Line 199: Soil DNA was extracted using a FastDNA Spin kit (MP Biomedicals, USA).*

**Comment # 15:**

Line 233: change 'agricultural managements' to 'agricultural management types'

*Response: Thank you for your suggestion. We have revised the sentence.*

*Line 233-235: Structural equation modeling (SEM) was used to explore the relationships among agricultural management types, soil properties, and P-cyclingrelated genes by Amos (24.0).*

**Comment # 16:**

Line 238: change 'incorporated' to 'in combination with'

*Response: Thank you for your suggestion. We have revised it as suggested.*

*Line 239: Straw retention in combination with mineral fertilization (i.e., W0M1F1, W1M0F1, W1M1F1) decreased soil pH from 6.90 to 5.01.*

**Comment # 17:**

Line 239: better mention the values changed from pH x to y than the differences

*Response: Thank you for pointing that out. We have revised it as suggested.*

*Line 239: Straw retention in combination with mineral fertilization (i.e., W0M1F1, W1M0F1, W1M1F1) decreased soil pH from 6.90 to 5.01.*

**Comment # 18:**

Line 240 and throughout the manuscript: use SI-units e.g. g kg-1 or µg g-1 h-1

*Response: Thank you for your feedback. We have revised the units throughout the manuscript.*

**Comment # 19:**

Line 243: change to: there was not significant effect of straw retention detectable

*Response: Thank you for your suggestion. We have revised it as suggested.*

*Line 246-248: There was not significant effect of sole straw retention (i.e., W1M1F0) detectable except for slight increases in soil MBC and MBP contents compared with the control treatment (Table 1).*

**Comment # 20:**

Line 247-250: this is very vague, can this turned more specific maybe?

*Response: Thank you for pointing that out. We have revised the descriptions.*

*Line 249-251: Mineral fertilization indeed enhanced soil nutrient contents. It also led to soil acidification, which was not effectively alleviated by the return of straw in combination with mineral fertilization.*

**Comment # 21:**

Line 279: remove 'that' after manifested

*Response: Thank you for your suggestion. We have removed 'that' after manifested in Line 283.*

*Line 282-283: These results manifested the effect of mineral fertilization on P species concentration was more apparent than that of straw retention.*

**Comment # 22:**

Line 283: remove 'the' before samples

*Response: Thank you for your suggestion. We have removed 'the' before samples in Line 287.*

*Line 284-286: Notably, the concentrations of orthophosphate, orthophosphate monoesters, orthophosphate diesters, and Glyc+nucl (i.e., α/β-glycerophosphate and mononucleotides) in WECs were significantly greater (~2.5 times) than those in bulk soil for all tested samples (Tables 4 and 5).*

**Comment # 23:**

Line 286: stick to uniform tenses

*Response: Thank you for pointing that out. We have revised it as suggested.*

*Line 287-290: Relative to the control, the concentrations of orthophosphate, orthophosphate monoesters and orthophosphate diesters rose sharply after mineral fertilization for WECs, while the significant increase of only orthophosphate concentrations was detected for bulk soils.*

**Comment # 24:**

Line 316: remove 'the' before treatments

*Response: Thank you for your feedback. We have removed 'the' before treatments in Line 320.*

*Line 318-319: Additionally, the bacterial β-diversity in WECs showed a clear divergence from those in bulk soils for all treatments (Fig. S7).*

**Comment # 25:**

Line 322: this is not a full sentence, please modify (whereas at the begin of the sentence needs to be changed).

*Response: Thank you for your suggestions. We have revised it as suggested.*

*Line 323-325: Specially, a strong correlation was detected between the majority of P cycling genes and soil nutrients including C, N, P in bulk soils. In contrast, no consistent trends were observed in WECs.*

**Comment # 26:**

Line 390: ... I suggest to change to: could be caused by the higher specific surface area of the WEC

*Response: Thank you for pointing that out. We have revised it as suggested.*

*Line 392-394: The higher concentrations of SOC, TN, TP, AP and various P species in WECs (Fig. 1 and Table 5) compared with bulk soil (Tables 1 and 4) indicated nutrient enrichment within the WECs. This could be caused by the higher specific surface area of the WECs (Jiang et al., 2014).*

**Comment # 27:**

Line 391 and general, some sentences are very long and could be shortened

*Response: Thank you for your feedback. We have revised the sentences.*

*Line 394-396: Significant increases in these indicators suggested that the management practices exerted more substantial impacts on soil properties and P species in WECs than in bulk soils.*